# Learning Physical Simulation with Historical Message-Passing Integration Transformer

## Abstract

Machine learning methods for mesh-based physical simulation have achieved significant success in recent years. (Reviewer ZnqH) We propose the Historical Message-Passing Integration Transformer (HMIT), an architecture based on Graph Neural Networks that incorporates a message passing framework and applies Graph Fourier Loss (GFL) for model optimization. (Reviewer ZnqH) To mitigate over-squashing, capture fine-grained details, and scale linearly with node count, we introduce Historical Message-Passing Attention (HMPA), which integrates multi-step historical message-passing information for each node with feature-wise softmax and employs a decoder-only architecture. Additionally, to modulate loss at specific frequencies and handle varying energy levels, we introduce GFL, which uses a frequency-domain energy adjustment schedule. To improve computational efficiency, we precompute the graph's Laplacian eigenvectors before training. Our architecture achieves significant accuracy improvements in shart- and long-term rollouts for both Lagrangian and Eulerian dynamical systems compared to current methods.

## 1 Introduction

In recent advancements in physical systems simulation, an increasing number of neural network-based methods are challenging traditional numerical solvers. These methods, notable for being several orders of magnitude faster and maintaining low error rates, have garnered substantial attention from researchers (1; 2; 3; 4). Graph Neural Networks (GNNs) have garnered widespread interest and research attention due to their unique features, including node-wise independent updates and aggregation operations that closely resemble iterations in traditional simulations. Additionally, particle-based and mesh-based simulations can be easily converted into graphs through neighbor interactions (4) and topology structures (5). This growing body of research can be collectively termed as GNN for Simulation (GNN4Sim) (6; 7; 8; 9).

Unlike the fields of Natural Language Processing (NLP) and Computer Vision (CV), physical systems are more complex and unstable, where every piece of information is crucial. This implies that issues like over-smoothing and over-squashing might be more pertinent in physical simulations (10). A common solution to mitigate these issues is mapping the original information to a high-dimensional latent space for processing. This approach has become a widely used architecture in most GNN4Sim applications (11; 5; 12; 4; 13), known as the encoder-processor-decoder (EPD), which, due to its focus on nodes and edges rather than graph structure, has a broad range of applications.

Since the Attention mechanism achieved significant success in the Natural Language Processing (NLP) domain (14), many Transformer-based network architectures have been developed in computer vision (15; 16), Social and Information Networks (17), (Reviewer ZnqH, Reviewer L6SR, Reviewer G8Yr) and physical simulation (18; 19; 20). Universal architectures for graph-based learning have also emerged, such as GraphGPS (21), Graph Attention Networks (GAT) (22), and Graph MLP-Mixer (23). However, physical simulations pose unique challenges that require tailored solutions.

To address these challenges, we propose **Historical Message-Passing Attention (HMPA)**, a novel mechanism designed to solve three key issues commonly encountered in physical simulations:

1. **Mitigating over-squashing**: Over-squashing occurs when important information from distant nodes fails to propagate effectively (24), leading to poor long-range dependencies. To address this, HMPA integrates *multi-step historical message-passing information* for each node, enabling nodes to aggregate and retain context from earlier message-passing steps. This facilitates the effective propagation of critical information across varying distances.

2. **Capturing fine-grained details**: Physical simulations often involve intricate dynamics that require precise modeling of feature importance (10). HMPA applies *feature-dimensional softmax combined with Hadamard products*, which assigns attention weights to individual feature dimensions rather than sequence positions. This fine-grained weighting enhances the model's ability to focus on the most relevant features for each node, improving accuracy.

3. **Scaling linearly with node count**: Standard self-attention mechanisms suffer from quadratic complexity with respect to the number of nodes, limiting their applicability to large-scale systems. HMPA adopts a *decoder-only design*, which ensures linear scalability with the number of nodes, making it efficient and suitable for large physical simulations.

Incorporating theoretical methodologies from graph signal processing, we have innovatively applied the Graph Fourier Transform (GFT) (25; 26) to the domain of physical simulations through our introduction of Graph Fourier Loss (GFL). This novel loss function optimizes model performance by leveraging the unique spectral properties of graphs. Rather than applying a Fourier Transform directly to the model, we employ preprocessing Laplacian eigenvector matrix within the loss function, thereby keeping the model's inference time nearly unchanged.

## 2 RELATED WORK

**Neural Approaches in Physical Systems Simulation**    In recent years, Several neural-based methods have achieved great research interest in physical systems simulation due to their high computational speed and low error rates. Among these, Convolutional Neural Networks (CNNs) (27; 28; 29; 30; 31; 32) have been utilized to infer the dynamics of physical systems, demonstrating the capability of neural networks in accurately and efficiently simulating complex phenomena. Physics-Informed Neural Networks (PINNs) (3; 33; 34; 35; 36) , which leverage implicit representations and physical condition constraints, can be trained without traditional datasets, illustrating a groundbreaking approach to model training that is particularly valuable in scenarios where empirical data is scarce or difficult to obtain. Neural operators (2; 37; 38; 39) offer a novel methodology for predicting the physical state at any given time step directly from initial conditions. This represents a significant shift from traditional simulation methods, enabling more efficient and flexible simulations across various scales and conditions. Graph Neural Networks (GNNs) (4; 5; 40; 41) facilitate information exchange between nodes through message passing. This mechanism effectively captures the interactions within physical systems, allowing for the detailed simulation of complex dynamics. Closely related is differentiable simulation, which emerge as a powerful tool to address optimization applications in different applications, including fluids (42), cloth (43; 44), deformable objects (45), articulated bodies (46), and solid-fluid coupling systems (47; 48; 49).

**Advancements in GNN Architectures for Simulation**    Recent studies in GNN4Sim have introduced various improvements. Multiscale methods (50; 51; 52; 53) , benefiting from simplified latent graph structures, have significantly accelerated training and inference while maintaining quality. At its core, this involves enhancing the efficiency of message passing, which entails modifying the topological structure of the graph. This change aims to concurrently increase processing speed and enhance accuracy in the simulations. Similarly, FIGNet (54) by adding face-face edges, changes the graph structure to improve collision accuracy. Han et al. (55) through uniform sampling, simplifies the graph structure, applying scaled dot-product attention to the entire graph but requires multiple prior temporal steps information. TIE (56) streamlines interaction modeling in Message Passing Neural Networks, utilizing a modified attention mechanism to efficiently process particle dynamics without explicit edge representations. LAMP (57) uses reinforcement learning to adapt to the varying relative importance of the trade-off between error and computation at inference time. C-GNS (58) focus on model the constraints of the physical system.

**Bridging Graph Theory and Signal Processing**    Graph Signal Processing (GSP) extends traditional signal processing techniques to signals defined on graphs (59; 60) , a paradigm shift that has

unlocked new avenues in analyzing complex data structures. Bruna et al. (61) introducing Spectral Networks for graph data learning, establishing foundational techniques for GNNs. Sandryhaila and Moura (25) and Hammond et al. (26) introduced the concept of applying wavelet transforms on graphs, offering a powerful tool for signal analysis and processing on irregular domains. Further, the development of Graph Convolutional Networks (GCNs) (62; 63; 64) , simplified the application of convolutional neural networks to graph data, enabling efficient learning of graph-structured data. These foundational studies emphasize the significance of spectral methods in understanding and leveraging the inherent structure of data represented as graphs.

# 3 PROBLEM FORMULATION AND PRELIMINARIES

This section introduces the problem formulation and the necessary preliminary concepts. It begins with the representation of physical systems using graph structures and the optimization goal for a learnable simulator in Section 3.1. It then delves into the Graph Fourier Transform (GFT), which facilitates the analysis of graph signals in the spectral domain in Section 3.2.

## 3.1 PROBLEM FORMULATION

(Reviewer G8Yr) We consider graph $G^t = (V^t, E^t)$ to represent a physical system with $t$ taking discrete values $t = 0, 1, \ldots$, where $V^t$ denotes the set of nodes with node attributes $v_i^t$ for each $v_i^t \in V^t$, and $E^t$ denotes the set of edges with edge attributes $e_{ij}^t$ for each $e_{ij}^t \in E^t$. We also define a total of $M$ Message Passing iterations, with $k = 0, 1, \ldots, M$. During the $k$-th Message Passing iteration, the attributes of nodes and edges are denoted by $v_{k,i}^t$ and $e_{k,ij}^t$.

The learnable simulator $f_\theta$, parameterized by $\theta$, can be optimized towards training objective. The goal of the learnable simulator is to predict the next state of the system, $G^{t+1}$, based on the previous prediction of graph $G^t$ at time step $t$, denoted by $G^{t+1} = f_\theta(G^t)$, or $G^0 \rightarrow G^1 \rightarrow \cdots \rightarrow G^t$.

## 3.2 GRAPH FOURIER TRANSFORM

The Graph Fourier Transform (GFT) transforms signals on a graph from the spatial vertex domain to the spectral frequency domain. For a signal defined on the vertices of the graph, GFT leverages the eigenvectors of the graph's Laplacian matrix, projecting the signal onto the orthogonal basis formed by these eigenvectors. This projection allows us to analyze and process the signal in a domain where convolution and filtering can be performed algebraically.

### 3.2.1 MATHEMATICAL DEFINITIONS

(Reviewer wc4g) The adjacency matrix of $G$ is denoted by $A$, where $A_{ij} = 1$ if there is an edge between vertices $i$ and $j$, and $A_{ij} = 0$ otherwise. The degree matrix $D$ is a diagonal matrix where $D_{ii} = \sum_j A_{ij}$. The Laplacian matrix of the graph is defined as $L = D - A$.

The eigenvalues and eigenvectors of $L$ are denoted by $\lambda_i$ and $u_i$, respectively, where $i = 1, 2, \ldots, N$, and $N$ is the number of vertices in the graph.

Given a signal $x \in \mathbb{R}^N$ defined on the vertices of the graph, the GFT of $x$ is given by

$$\hat{x} = U^T x$$

where $U = [u_1, u_2, \ldots, u_N]$ is the matrix of eigenvectors of $L$, and $U^T$ is its transpose. The signal $x$ can be reconstructed from its GFT $\hat{x}$ using the inverse GFT, given by

$$x = U\hat{x}$$

# 4 METHOD

In Section 4.1, we present the overall architecture of the model, followed by a detailed description of the Historical Message-Passing Attention and Graph Fourier Loss in Sections 4.2 and 4.3, respectively. The Historical Message-Passing Attention is introduced to address aggregation bias and enable more fine-grained feature processing. The Graph Fourier Loss is introduced to balance the high-energy and low-energy components in the spectral domain, thereby enhancing the model's capacity to learn complex physical phenomena.

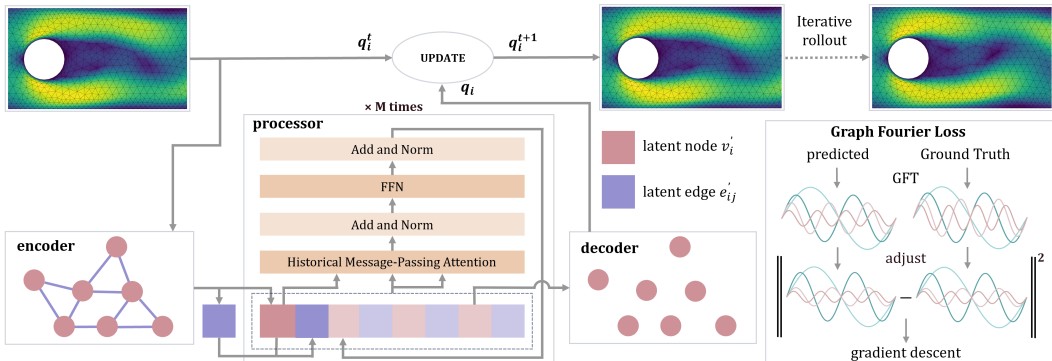

Figure 1: Model Architecture of the Historical Message-Passing Integration Transformer, visualizing the information processing procedure for the first of four Message Passing ($k = 0, M = 4$) times. The encoder module transposes inputs into a latent space and the decoder predicts future states by extrapolating these encoded representations. The processor unit conducts numerous iterations, each treated as a regression problem, to refine node and edge attributes. The highly complex physical details make the model sensitive to noise, so the dynamic modulation of the frequency domain energy using Graph Fourier Loss (GFL) attenuates the impact of noise. GFL leverages the spectral properties of graphs to enhance model inference efficacy.

## 4.1 HISTORICAL MESSAGE-PASSING INTEGRATION TRANSFORMER

The Historical Message-Passing Integration Transformer architecture incorporates a Message Passing framework, employs an Encoder-Processor-Decoder structure, and utilizes Graph Fourier Loss for model optimization. Figure. 1 visualizes the computational process of the model.

**Encoder**   The node and edge attributes are transformed into a latent space by $f_1$ and $f_2$, respectively.

$$v_{0,i}^t \leftarrow f_1(v_i^t), \quad e_{0,ij}^t \leftarrow f_2(e_{ij}^t)$$

**Processor**   The edge features are updated by $f_3$, incorporating features from adjacent nodes. Node features are then updated by $f_4$, which aggregates information across multiple tokens using Historical Message-Passing Attention. Each token represents node and aggregated edge features from a particular Message Passing iteration:

$$e_{k+1,ij}^t \leftarrow f_3(e_{k,ij}^t, v_{k,i}^t, v_{k,j}^t), \quad v_{k+1,i}^t \leftarrow f_4\left(v_{k,i}^t, \bigoplus_{m=0}^{k}\left(v_{m,i}^t, \sum_j e_{m,ij}^t\right)\right)$$

where $\bigoplus$ denotes the sequential concatenation of tokens $\left(v_{m,i}^t, \sum_j e_{m,ij}^t\right)$ for each $m$ from 0 to $k$, forming the input sequence for $f_4$.

We choose to use historical Message Passing (MP) steps as sequence inputs instead of traditional temporal features because our model focuses on single-step prediction—using information at time $t$ to predict the state at time $t+1$. Different MP steps capture various aspects of node states, enriching the feature representation at each iteration. This approach aligns with models like MeshGraphNet (5), which utilize independent MLPs for each MP step due to the unique information each step provides. (Reviewer ZnqH) By leveraging historical MP step features as context for the current MP step, we mitigate over-squashing by enabling the model to propagate information across multiple MP steps, capturing broader dependencies in complex physical environments. The processor treats node updates as an autoregressive problem, using past message-passing attributes for keys ($K$) and values ($V$) and the current state as the query ($Q$). (Reviewer ZnqH) This decoder-only design ensures efficient information aggregation, scaling linearly with the number of nodes, while preserving critical features for robust predictions.

**Decoder** After $M$ times Message Passing, the latent node features are mapped back to the original attribute space by $f_5$, culminating in the update of the graph state to the next time step.

$$v_i^{t+1} \leftarrow f_5(v_{M,i}^t), \quad G^{t+1} = \text{UPDATE}\left(G^t, v_i^{t+1}\right)$$

Here, $f_1$, $f_2$, $f_3$, and $f_5$ are all shallow MLPs. During training, finally, we compute the model loss using our Graph Fourier Loss and update the weights accordingly.

## 4.2 Scaled Historical Message-Passing Attention

To address effectively sidestepping the aggregation bias introduced by the summation operations typical of matrix multiplication, we introduce the Scaled Historical Message-Passing Attention (HMPA) mechanism. HMPA utilizes element-wise multiplication followed by a linear transformation to finalize the attention computation, (Reviewer ZnqH) enabling the model to capture fine-grained details by focusing on the relative importance of individual feature dimensions. This design ensures that nuanced information is preserved and emphasized during message-passing updates, leading to more accurate and expressive representations.

Our attention mechanism is tailored for inputs with a finite maximum sequence length, enabling more nuanced processing of the relative importance of features within the message passing framework of the processor. Let $v_{k,i} \in \mathbb{R}^d$ denote the feature vector of the current node attribute at iteration $k$ for node $i$, and $\sum_j e_{k,ij} \in \mathbb{R}^d$ represent the aggregated edge attributes associated with node $i$ at iteration $k$. We construct the key and value matrices $K, V \in \mathbb{R}^{s \times d}$ by concatenating the sequences of node attributes and their corresponding aggregated edge attributes from iterations $0$ to $k$:

$$K = V = \bigoplus_{m=0}^{k} \left(v_{m,i}, \sum_j e_{m,ij}\right) = [v_{0,i}, \sum_j e_{0,ij}, \ldots, v_{k,i}, \sum_j e_{k,ij}]$$

Here, $s = 2(k+1)$ is the sequence length, and $d$ is the feature dimension.

The corresponding attention weights $a \in \mathbb{R}^{s \times d}$ are computed by applying a scaled Hadamard product between the current node attribute $v$ and the key matrix $K$, followed by a softmax operation along the feature dimension:

$$a = \text{softmax}\left(\frac{v \odot K}{\sqrt{d}}\right)$$

where $\odot$ denotes element-wise multiplication between $v$ (broadcasted to match the dimensions of $K$) and $K$, and the softmax function is applied over the feature dimension $d$ for each sequence position. Consequently, the contribution of each dimension to the Value vector's computation is determined by its relative importance across the dimension, not by its position within the sequence. Specifically, the element-by-element representation of matrix $a$ is:

$$a_{p,q} = \frac{\exp\left(\frac{v_q \cdot K_{p,q}}{\sqrt{d}}\right)}{\sum_{q'=1}^{d} \exp\left(\frac{v_{q'} \cdot K_{p,q'}}{\sqrt{d}}\right)}$$

Subsequently, the attention weights $a$ are applied to the value matrix $V$ through element-wise multiplication, yielding the weighted value matrix $w \in \mathbb{R}^{s \times d}$, which undergoes a linear transformation to reshape it back to a dimension of $d$.

Our methodological shift fundamentally reorients the attention mechanism from focusing on sequence positions to emphasizing feature dimensions. Unlike the traditional Scaled Dot-Product Attention, which assigns scalar attention weights to each position in the sequence, our Scaled Historical Message-Passing Attention allocates weights across each feature dimension. By applying the softmax function along the feature dimension $d$ rather than the sequence dimension $s$, HMPA capture the relative importance of individual features in contributing to the node updates. Channel mixing is then performed by the linear transformation.

For cases requiring multiple attention heads, we extend HMPA to its multihead version, Multihead Historical Message-Passing Attention, in a manner similar to the multihead extension of Scaled Dot-Product Attention, allowing the model to attend to different feature dimensions simultaneously.

### 4.2.1 COMPLEXITY ANALYSIS

(Reviewer YBDW) The floating-point operations (FLOPs) for the HMPA mechanism are calculated with $N$ representing the number of nodes, $s$ as the sequence length, and $d$ as the feature dimension. Constructing the key and value matrices $K$ and $V$ requires linear transformations across each sequence step, amounting to $2Nsd^2$ FLOPs. The calculation of attention weights $a$, including the element-wise product and softmax operation, requires $2Nsd$ FLOPs. The element-wise multiplication of $a$ and $V$, followed by a linear transformation to reshape the result back to dimension $d$, adds $Nsd + Nsd^2$ FLOPs. Summing these components, the total FLOPs for HMPA is:

$$\text{FLOPs} = 3Nsd^2 + 3Nsd.$$

This complexity scales **linearly** with the number of nodes $N$, with a fixed sequence length $s$. Compared to a global self-attention mechanism, our decoder-only architecture avoids the $O(N^2)$ complexity, making it especially efficient and advantageous for large-scale simulations.

### 4.2.2 SELECTIVE FEATURE AGGREGATION

HMPA enhances the message-passing process by concentrating on feature-level adjustments, avoiding the homogenization of information often seen in traditional attention mechanisms that aggregate over sequence positions. The element-wise multiplication and feature-dimension-specific softmax prevent less informative features from overshadowing crucial ones, maintaining the unique contributions of each feature to the node updates. As a result, nodes can selectively aggregate the most relevant features from their neighbors, leading to richer and more discriminative node embeddings that better capture the underlying graph structure.

### 4.3 GRAPH FOURIER LOSS

While the Scaled Historical Message-Passing Attention (HMPA) mechanism enhances the model's ability to focus on critical feature dimensions during message passing, it is equally important to ensure that the learned representations capture the essential spectral properties of the graph data. To this end, we introduce the **Graph Fourier Loss (GFL)**, which complements HMPA by promoting a balanced learning of both high-energy and low-energy components in the spectral domain. Together, HMPA and GFL jointly optimize the model's performance by addressing feature importance in both the spatial and spectral domains.

**Preprocessing** When the model does not alter the graph's topological structure, the inherent topological properties of the graph, such as the Laplacian matrix and its eigenvalues and eigenvectors, remain unaltered throughout the training process. To avoid the substantial increase in computation time caused by calculating eigendecompositions in each forward pass of our model, we preprocess the training set before commencing model training. For each time step, the graph's Laplacian matrix is calculated and subsequently decomposed into eigenvectors $U$. Consequently, during training, we only need to call the eigenvectors to calculate the loss, and during inference, the eigenvectors are not required at all.

**Compute Graph Fourier Loss during training** To circumvent the significant computational overhead of calculating eigenvectors during inference, we propose the Graph Fourier Loss (GFL) as the loss function. This strategy ensures the inference speed of the model remains unaffected.

Initially, we perform GFT on both the model's output $y^{\text{train}} \in \mathbb{R}^{N \times d}$ and the target output $y \in \mathbb{R}^{N \times d}$, transforming the signals from the time domain to the frequency domain:

$$\hat{y} = U^T y, \quad \hat{y}^{\text{train}} = U^T y^{\text{train}}$$

Subsequently, we calculate the energy of each dimension of the transformed signals and sum them up to obtain the total energy for each signal across all nodes and dimensions:

$$E = \sum_{k=1}^{d} |\hat{y}_{:,k}|^2, \quad E_{\text{train}} = \sum_{k=1}^{d} |\hat{y}_{:,k}^{\text{train}}|^2$$

$E$ and $E_{\text{train}}$ represent the total energy of the target and model output signals in the frequency domain, respectively.

The energy $E$ are then sorted, and using the hyperparameter segment rate $s_r$, it is divided into high $E_{\text{high}}$ and low $E_{\text{low}}$ energy components. An adjustment factor $\alpha$ is computed based on the mean energy of these partitions:

$$\alpha = \sqrt{\frac{\text{mean}(E_{\text{high}})}{\text{mean}(E_{\text{low}}) + \epsilon}} \cdot \lambda$$

The constant $\epsilon$ is employed to prevent division by zero, while the regularization parameter $\lambda$ controls the strength of the adjustment. When $\alpha > 1$, high-energy regions are amplified, resulting in the model emphasizing high-energy components. Conversely, when $\alpha < 1$, this emphasis is reduced. Both manual setting of the regularization parameter $\lambda$ and incorporation it as a learnable parameter have been tested, found in Experiment 5.4

Finally, we adjust the signals and compute the mean squared error (MSE) directly in the spectral domain:

$$\hat{y}' = \text{adjust}(\hat{y}, \alpha), \quad \hat{y}^{\text{train}'} = \text{adjust}(\hat{y}^{\text{train}}, \alpha),$$

where the adjust($\cdot$) function operates on the spectral signals and scales their low-energy components by $\alpha$, leaving high-energy components unchanged:

$$\text{adjust}(\hat{y}, \alpha)_i = \begin{cases} \alpha \cdot \hat{y}_i, & \text{if } i \in \text{low-energy components}, \\ \hat{y}_i, & \text{if } i \in \text{high-energy components}. \end{cases}$$

The Graph Fourier Loss is then defined as:

$$\text{GFL} = \frac{1}{N} \|\hat{y}' - \hat{y}^{\text{train}'}\|_2^2.$$

By integrating GFL with HMPA, the model effectively captures essential information in both the spatial and spectral domains, leading to improved predictive performance in complex physical environments. In Appendix A, we analyze the gradient with respect to $\lambda$ and explain why $\lambda$ does not converge to zero. The presence of both positive and negative terms in the derivative suggests the existence of an optimal $\lambda > 0$ that minimizes the loss. In Appendix C, we provide a theoretical analysis of why GFL is effective. The adjustment factor $\alpha$ serves as a frequency-specific weight, modulating the importance of each frequency component. Additionally, by incorporating $\frac{\partial \hat{y}_i}{\partial \theta}$, the model integrates frequency domain information, improving its ability to capture meaningful patterns across frequencies. The interaction between $\alpha$ and the error terms ensures an adaptive learning process that shifts focus towards the most relevant frequency components.

## 5 EXPERIMENTS

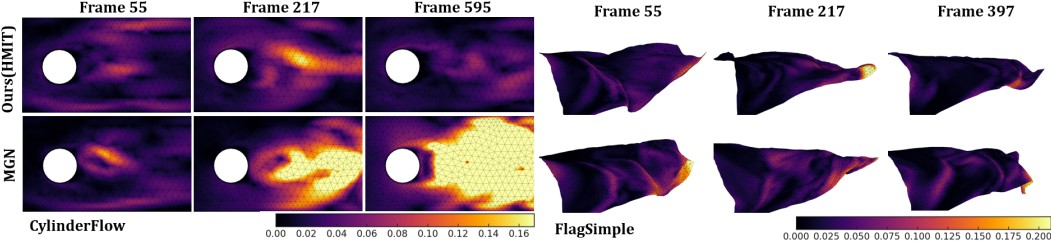

Figure 2: Comparison of RMSE of velocity norm between the Lagrangian system *FlagSimple* and the Eulerian system *CylinderFlow* using our HMIT and MeshGraphNet (MGN) (5).

In Section 5.1, we describe the datasets and implementation details, followed by an analysis of the precomputation costs associated with GFL in Section 5.3. Subsequently, we present the baseline models used for comparison and discuss the evaluation results, along with ablation studies conducted to assess the contributions of specific model components in Sections 5.2 and 5.4. We also visualize the RMSE of a Lagrangian system and an Eulerian system respectively, as shown in Figure 2.

## 5.1 TASK SETUPS

**Datasets Description**   We evaluated our method in the representation of both Lagrangian and Eulerian dynamical systems. The Lagrangian systems involve the datasets *FlagSimple* and *DeformingPlate*, while the Eulerian systems include *CylinderFlow* and *Airfoil*, with all datasets sourced from MeshGraphNet (5).

- *FlagSimple* models a flag blowing in the wind, utilizing a static Lagrangian mesh with a static topology structure and ignores collisions.
- *DeformingPlate* Utilizes a quasi-static simulator to model the deformation of a hyperelastic plate by a kinematic actuator. The dataset is structured with a Lagrangian tetrahedral mesh.
- *CylinderFlow* simulates the flow of an incompressible fluid around a fixed cylinder in a 2D Eulerian mesh.
- *Airfoil* focuses on the aerodynamics around an airfoil wing section, employing a 2D Eulerian mesh to monitor the evolution of momentum and density.

**Implementation**   Our framework is built using PyTorch (65) and PyG (PyTorch Geometric) (66). The entire model is trained and inferred on a single Nvidia RTX 4090. Detailed information, including network hyperparameters, input and output formats, and noise injection methods, can be found in appendix E. Our datasets and code are publicly available at https://github.com/Heiyanyan/LearningPhysical-Simulation-with-Message-Passing-Transformer.

| Measurements | Dataset | HMIT (ours) | MGN (5) | BSMS (50) | TIE (56) | Mesh Transformer (67) | Graph MLP-Mixer (23) |
|---|---|---|---|---|---|---|---|
| RMSE-1 [1E-2] | Cylinder | **2.03E-01** | 5.24E-01 | 5.09E-01 | 4.21E-01 | 3.05E-01 | 4.12E-01 |
| | Airfoil | **2.61E+02** | 3.14E+02 | 2.94E+02 | 3.17E+02 | 2.96E+02 | 3.05E+02 |
| | Plate | **1.00E-02** | 2.69E-02 | 2.83E-02 | 3.56E-02 | 2.38E-02 | 3.28E-02 |
| | Flag | **1.12E-02** | 6.47E-02 | 6.51E-02 | 5.48E-02 | 4.73E-02 | 6.89E-02 |
| RMSE-50 [1E-2] | Cylinder | **6.32E-01** | 1.40 | 3.25 | 6.85 | 1.07 | 4.61 |
| | Airfoil | **4.08E+02** | 5.36E+02 | 1.34E+03 | 5.72E+03 | 5.21E+02 | 6.32E+02 |
| | Plate | **9.25E-02** | 1.73E-01 | 2.81E-01 | 3.61E-01 | 1.30E-01 | 6.19E-01 |
| | Flag | **1.87** | 2.29 | 2.46 | 2.19 | 2.04 | 3.90 |
| RMSE-all [1E-2] | Cylinder | **3.78** | 4.32 | 1.36E+01 | 2.68E+01 | 4.26 | 2.05E+01 |
| | Airfoil | **1.64E+03** | 2.08E+03 | 1.01E+04 | 1.27E+05 | 2.00E+03 | 3.97E+03 |
| | Plate | **1.09** | 1.61 | 4.52 | 9.62 | 1.28 | 8.22 |
| | Flag | **2.05** | 2.45 | 3.28 | 1.24E+01 | 2.32 | 7.25 |

Table 1: RMSE of our method, MeshGraphNet (MGN), Bi-Stride Multi-Scale GNN (BSMS-GNN), Transformer with Implicit Edges (TIE), Mesh Transformer and Graph MLP-Mixer for different rollout steps. Our method achieves state-of-the-art in all datasets.

## 5.2 COMPARISON WITH BASELINES

**Baselines**   In our evaluation, we compared against several state-of-the-art GNNs. The Bi-Stride Multi-Scale Graph Neural Network (BSMS) (50) introduces multiscale methods to enhance the efficiency of message passing. MeshGraphNet (MGN) (5) leverages a mesh-based approach for graph representation. The Transformer with Implicit Edges (TIE) model (56) streamlines interaction modeling in Message Passing Neural Networks by utilizing a modified attention mechanism to efficiently process particle dynamics without explicit edge representations. (Reviewer L6SR, Reviewer YBDW, Reviewer ZnqH) Mesh Transformer (67) incorporates global attention and hierarchical pooling mechanisms to capture long-range dependencies on non-uniform meshes. Graph MLP-Mixer (23) uses Hadamard-Product Attention between local patch encodings.

**Evaluation**   Table 1 demonstrates the superiority of our model across all datasets. We randomly selected three seeds to initialize the models and reported the mean RMSE values with their respective variance in the table. The *CylinderFlow* dataset at RMSE-1 reveals a pronounced improvement with our model, which shows a reduction in error by **33.4%** compared to Mesh Transformer, the nearest competitor. At RMSE-50 and RMSE-all, our model continues to exhibit superior performance,

showing a reduction in error by **40.9%** at RMSE-50 and by **11.3%** at RMSE-all when compared to Mesh Transformer.

In the context of the *Airfoil* dataset, our model remains state-of-the-art. At the RMSE-50 condition, the model's error rate is reduced by **21.6%** compared to Mesh Transformer. This illustrates the model's capacity to maintain accuracy over prolonged sequences, which is an essential feature for simulations requiring stability over extended temporal spans. At RMSE-all, the improvement reaches **18%**.

For the *DeformingPlate* and *FlagSimple* datasets, our model displays similar trends. In the Plate dataset, the RMSE-1 shows an improvement of **57.9%** over Mesh Transformer, with continued dominance in longer simulations, indicated by a **28.8%** error reduction at RMSE-50. For the FlagSimple dataset, while the improvements are more significant, our model consistently outperforms other methods across all metrics, with the most notable reduction being **76.7%** at RMSE-1.

## 5.3 PRECOMPUTATION COST

| Dataset | Eigen Time per Sample (s) | Total Time (s) |
|---|---|---|
| Cylinder | 0.026 | 27.21 |
| Airfoil | 0.204 | 218.34 |
| Plate | 0.016 | 19.19 |
| Flag | 0.020 | 21.82 |

Table 2: Preprocessing costs for Graph Laplacian eigen decomposition.

(Reviewer G8Yr, Reviewer L6SR, Reviewer wc4g, Reviewer YBDW, Reviewer ZnqH) To provide clarity on preprocessing costs, we evaluate both the eigen decomposition time for the Graph Laplacian and the total preprocessing time across datasets. These computations only occur once and are considered part of the dataset generation process. For static graph topologies, eigen decomposition is only performed at $t = 0$. For datasets with dynamic topologies, eigen decomposition is conducted at each time step. This approach eliminates the computational complexity that would otherwise be incurred during training, while ensuring that the inference speed remains unaffected. The maximum preprocessing time across all datasets is only 3 minutes, which is negligible compared to the training duration.

## 5.4 ABLATION STUDIES

| Measurements | Dataset | Without HMPA and GFL | HMPA only | GFL only | HMPA + GFL (ours) |
|---|---|---|---|---|---|
| RMSE-1 [1e-2] | Cylinder | 5.83E-1 | 2.64E-1 | 2.27E-1 | **2.03E-1** |
| | Flag | 6.47E-2 | 1.51E-2 | 2.29E-2 | **1.12E-2** |
| RMSE-50 [1e-2] | Cylinder | 1.42 | 9.10E-1 | 6.96E-1 | **6.32E-1** |
| | Flag | 2.29 | 1.97 | 2.03 | **1.87** |
| RMSE-all [1e-2] | Cylinder | 4.32 | 3.94 | 3.89 | **3.78** |
| | Flag | 2.45 | 2.16 | 2.21 | **2.05** |

Table 3: Ablation study conducted on the CylinderFlow and (Reviewer wc4g) FlagSimple datasets to evaluate the contributions of individual components within our architecture. We test the effects of Historical Message-Passing Attention (HMPA), Graph Fourier Loss (GFL), and their combination. Results indicate that integrating both HMPA and GFL leads to reductions in error. (Reviewer ZnqH)When GFL is not used as the loss function, we replace it with MSE.

To rigorously evaluate the influence of specific model components and configurations on overall performance, systematic ablation studies were undertaken. These included: (1) Evaluating the contributions of individual components by comparing the effects of Historical Message-Passing Attention (HMPA) and Graph Fourier Loss (GFL), (2) comparing Dot-Product Attention with Historical Message-Passing Attention, (3) assessing the efficacy of learnable lambda parameters $\lambda$ versus man-

ual setting of $\lambda$, and (4) investigating how varying segmentation rates $s_r$ affect model performance. More experimental analysis of GFL can be found in Appendix B.

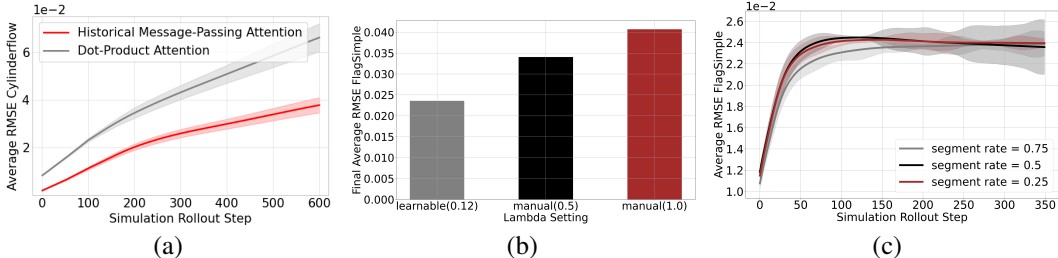

(a)           (b)           (c)

Figure 3: (a) Comparison of Dot-Product Attention and Historical Message-Passing Attention in *CylinderFlow*. HMPA demonstrates a lower average RMSE across all rollout steps compared to the Dot-Product Attention. (b) Comparison of learnable and manual $\lambda$ settings in *FlagSimple*. Learnable $\lambda$ achieves lower error compared to manual settings. (c) The impact of varying segmentation rates $s_r$ in *FlagSimple*. Different segmentation rates do not significantly impact the final results.

**Effectiveness of Graph Fourier Loss and Historical Message-Passing Attention**  We test the effects of our each component in Table 3. In terms of predictive accuracy, the model without HMPA and GFL performed the worst, demonstrating significantly higher error rates across all RMSE measures. The integration of both HMPA and GFL demonstrated the highest improvement in reducing error rates across all RMSE measures when compared to standalone implementations of GFL and HMPA.

**Effectiveness of Historical Message-Passing Attention**  Figure 3a compares the performance of Dot-Product Attention (14) and Historical Message-Passing Attention (HMPA) on the *CylinderFlow* dataset. HMPA consistently outperforms Dot-Product Attention across all rollout steps, demonstrating a lower average RMSE. This indicates that HMPA's finer-grained feature dimension weighting is more effective in capturing the dynamics of the system, leading to more accurate predictions.

**Effectiveness of Learnable $\lambda$**  Figure 3b compares the performance of models with learnable $\lambda$ settings against manual $\lambda$ settings in the *FlagSimple* dataset. The results show that the learnable $\lambda$ achieves a lower final average RMSE compared to manual settings. This highlights the advantage of allowing the model to adaptively adjust $\lambda$ during training, leading to better overall performance.

**Segmentation Rate Selection**  In Figure 3c, we investigate the impact of varying segmentation rates $s_r$ on the *FlagSimple* dataset. The results indicate that different segmentation rates do not significantly impact the final results. This robustness to segmentation rate selection demonstrates that our model can maintain high performance regardless of the specific value of $s_r$, simplifying the hyperparameter tuning process.

## 6 CONCLUSION AND LIMITATION

The Historical Message-Passing Integration Transformer (HMIT) has achieved notable advancements in the accuracy of physical system simulations by effectively integrating Historical Message-Passing Attention (HMPA) and Graph Fourier Loss (GFL). HMPA mitigates over-squashing, captures fine-grained details and scales linearly with node count, while GFL ensures the model's robustness by focusing on spectral balance. This synergy between HMPA and GFL results in a model that excels in long-term rollouts, providing accurate and reliable physical simulations. Continued development of HMIT could lead to broader applications in dynamic system modeling and enhance its utility in scientific and engineering fields, advancing the capabilities of learnable simulation technologies.

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

## A  ANALYSIS OF LEARNABLE $\lambda$ IN GRAPH FOURIER LOSS

Experiments revealed a notable phenomenon: when using a learnable $\lambda$ within the Graph Fourier Loss (GFL) framework, $\lambda$ did not tend to zero. Conversely, when applying a similar adjustment directly to the mean squared error (MSE) in the frequency domain, the value of $\lambda$ quickly diminished to zero. This appendix provides a comprehensive analysis of this observation and elucidates the underlying reasons.

The key difference between the GFL approach and direct MSE adjustment lies in the interaction of $\lambda$ with the frequency domain energy components. In GFL, $\lambda$ is indirectly involved through the calculation of an adjustment factor $\alpha$, which is applied separately to the model's output $y^{\text{train}}$ and the target output $y$. This can be expressed as:

$$
\text{GFL} = \frac{1}{N} \left\| \text{adjust}(U^\top y^{\text{train}}, \alpha) - \text{adjust}(U^\top y, \alpha) \right\|_2^2,
$$

where

$$
\text{adjust}(\hat{y}, \alpha)_i = \begin{cases} \alpha \cdot \hat{y}_i, & \text{if } i \in L_{\hat{y}}, \\ \hat{y}_i, & \text{if } i \in H_{\hat{y}}, \end{cases}
$$

and $L_{\hat{y}}$ and $H_{\hat{y}}$ denote the indices of the low- and high-energy components of $\hat{y}$, respectively.

The adjustment factor $\alpha$ is defined as:

$$
\alpha = \lambda \cdot \sqrt{\frac{\text{mean}(E_{\text{high}})}{\text{mean}(E_{\text{low}}) + \epsilon}},
$$

where $E_{\text{high}}$ and $E_{\text{low}}$ represent the energies of the high- and low-frequency components, respectively, and $\epsilon$ is a small constant to prevent division by zero. The parameter $\lambda$ helps balance the energy distribution across different frequency components, ensuring that it remains non-zero to maintain the desired balance between high- and low-frequency components.

### A.1  COMPUTATION OF $\frac{\partial \text{GFL}}{\partial \lambda}$

To understand why $\lambda$ does not tend to zero in GFL, the partial derivative of GFL with respect to $\lambda$ is computed. Denote:

$$
\hat{y}^{\text{adj}} = \text{adjust}(U^\top y^{\text{train}}, \alpha), \quad y^{\text{adj}} = \text{adjust}(U^\top y, \alpha),
$$

and define the error vector:

$$
e = \hat{y}^{\text{adj}} - y^{\text{adj}}.
$$

Then, GFL can be expressed as:

$$
\text{GFL} = \frac{1}{N} \|e\|_2^2 = \frac{1}{N} \sum_{i=1}^{N} e_i^2.
$$

Since the adjustment is applied separately to $y^{\text{train}}$ and $y$, and the division into low- and high-energy components may differ between them, different cases must be considered when computing the derivative.

### A.1.1  ADJUSTMENT CASES

Four cases are defined based on the indices of the components:

Case 1: $i \in L_{\hat{y}} \cap L_y$: Both adjusted as low-energy components.
$$
e_i = \alpha(\hat{y}_i - y_i).
$$

Case 2: $i \in L_{\hat{y}} \cap H_y$: Model output is low-energy, ground truth is high-energy.
$$
e_i = \alpha\hat{y}_i - y_i.
$$

Case 3: $i \in H_{\hat{y}} \cap L_y$: Model output is high-energy, ground truth is low-energy.
$$
e_i = \hat{y}_i - \alpha y_i.
$$

Case 4: $i \in H_{\hat{y}} \cap H_y$: Both are high-energy components.
$$
e_i = \hat{y}_i - y_i.
$$

### A.1.2 DERIVATIVE COMPUTATION

The derivative of $\alpha$ with respect to $\lambda$ is computed:

$$\frac{\partial \alpha}{\partial \lambda} = \sqrt{\frac{\mathrm{mean}(E_{\mathrm{high}})}{\mathrm{mean}(E_{\mathrm{low}}) + \epsilon}} = \frac{\alpha}{\lambda}.$$

Substituting the expressions for $e_i$ and $\frac{\partial e_i}{\partial \lambda}$, the total derivative can be expressed as:

Case 1:

$$\frac{\partial \mathrm{GFL}_1}{\partial \lambda} = \frac{2}{N} \sum_{i \in L_{\hat{y}} \cap L_y} (\alpha(\hat{y}_i - y_i)) \cdot \left( (\hat{y}_i - y_i) \cdot \frac{\partial \alpha}{\partial \lambda} \right) = \frac{2\alpha}{N} \frac{\partial \alpha}{\partial \lambda} \sum_{i \in L_{\hat{y}} \cap L_y} (\hat{y}_i - y_i)^2.$$

Case 2:

$$\frac{\partial \mathrm{GFL}_2}{\partial \lambda} = \frac{2}{N} \sum_{i \in L_{\hat{y}} \cap H_y} (\alpha \hat{y}_i - y_i) \cdot \left( \hat{y}_i \cdot \frac{\partial \alpha}{\partial \lambda} \right).$$

Case 3:

$$\frac{\partial \mathrm{GFL}_3}{\partial \lambda} = \frac{2}{N} \sum_{i \in H_{\hat{y}} \cap L_y} (\hat{y}_i - \alpha y_i) \cdot \left( -y_i \cdot \frac{\partial \alpha}{\partial \lambda} \right).$$

Case 4:

$$\frac{\partial \mathrm{GFL}_4}{\partial \lambda} = 0.$$

Combining all cases, the total derivative is:

$$\frac{\partial \mathrm{GFL}}{\partial \lambda} = \frac{2\alpha}{N} \frac{\partial \alpha}{\partial \lambda} \sum_{i \in L_{\hat{y}} \cap L_y} (\hat{y}_i - y_i)^2 + \frac{2}{N} \frac{\partial \alpha}{\partial \lambda} \left( \sum_{i \in L_{\hat{y}} \cap H_y} (\alpha \hat{y}_i - y_i)\hat{y}_i - \sum_{i \in H_{\hat{y}} \cap L_y} (\hat{y}_i - \alpha y_i)y_i \right).$$

Substituting $\frac{\partial \alpha}{\partial \lambda} = \frac{\alpha}{\lambda}$, the expression becomes:

$$\frac{\partial \mathrm{GFL}}{\partial \lambda} = \frac{2\alpha^2}{N\lambda} \left( \sum_{i \in L_{\hat{y}} \cap L_y} (\hat{y}_i - y_i)^2 + \sum_{i \in L_{\hat{y}} \cap H_y} (\alpha \hat{y}_i - y_i)\frac{\hat{y}_i}{\alpha} - \sum_{i \in H_{\hat{y}} \cap L_y} (\hat{y}_i - \alpha y_i)\frac{y_i}{\alpha} \right).$$

### A.1.3 ANALYSIS: WHY $\lambda$ DOES NOT TEND TO ZERO

The derivative $\frac{\partial \mathrm{GFL}}{\partial \lambda}$ indicates how changes in $\lambda$ affect the loss. The key observations are:

- **Balance of Frequency Components**: A non-zero $\lambda$ ensures that $\alpha$ adjusts the low-frequency components appropriately, maintaining a balance between high- and low-frequency energies.

- **Preventing Vanishing** $\alpha$: If $\lambda$ tends to zero, $\alpha$ also tends to zero, causing the adjusted low-frequency components to vanish. This would ignore important low-frequency information, degrading model performance.

- **Optimal** $\lambda$: The derivative includes both positive and negative terms due to the different cases. This suggests the existence of an optimal $\lambda > 0$ that minimizes the loss, rather than pushing $\lambda$ toward zero.

Therefore, during optimization, $\lambda$ is adjusted to balance the contribution of low-frequency components without diminishing them entirely.

## A.2 DIRECT APPLICATION OF $\lambda$ TO FREQUENCY DOMAIN MSE

Conversely, when $\lambda$ is directly applied to the MSE in the frequency domain using the combined error $y^{\text{train}} - y$, the lack of separate intermediate adjustments for $y^{\text{train}}$ and $y$ leads to a different effect. This can be expressed as:

$$\text{Adjusted MSE} = \frac{1}{N} \left\| \text{adjust}(U^\top(y^{\text{train}} - y), \alpha) \right\|_2^2.$$

In this formulation, the adjustment is applied after computing the error between the model output and the ground truth. The adjustment function modifies the error vector directly:

$$\text{adjust}(e, \alpha)_i = \begin{cases} \alpha \cdot e_i, & \text{if } i \in L_e, \\ e_i, & \text{if } i \in H_e, \end{cases}$$

where $L_e$ and $H_e$ denote the low- and high-energy components of the error vector $e = U^\top(y^{\text{train}} - y)$.

### A.2.1 COMPUTATION OF $\frac{\partial \text{ADJUSTED MSE}}{\partial \lambda}$

Since the adjustment is applied to the error vector as a whole, $\lambda$ affects the loss differently. The derivative is computed as:

$$\frac{\partial \text{Adjusted MSE}}{\partial \lambda} = \frac{2}{N} \sum_{i=1}^{N} \text{adjust}(e, \alpha)_i \cdot \frac{\partial \text{adjust}(e, \alpha)_i}{\partial \lambda}.$$

However, since $\alpha$ adjusts the error vector and $\alpha$ depends on $\lambda$, the derivative becomes:

$$\frac{\partial \text{adjust}(e, \alpha)_i}{\partial \lambda} = \begin{cases} e_i \cdot \frac{\partial \alpha}{\partial \lambda}, & \text{if } i \in L_e, \\ 0, & \text{if } i \in H_e. \end{cases}$$

Substituting $\frac{\partial \alpha}{\partial \lambda} = \frac{\alpha}{\lambda}$, the expression simplifies to:

$$\frac{\partial \text{Adjusted MSE}}{\partial \lambda} = \frac{2}{N} \sum_{i \in L_e} (\alpha e_i) \cdot \left( e_i \cdot \frac{\alpha}{\lambda} \right) = \frac{2\alpha^2}{N\lambda} \sum_{i \in L_e} e_i^2.$$

Since $e_i = \hat{y}_i - y_i$, the sum $\sum_{i \in L_e} e_i^2$ is always non-negative. Therefore, the derivative $\frac{\partial \text{Adjusted MSE}}{\partial \lambda}$ is non-negative.

### A.2.2 ANALYSIS: WHY $\lambda$ TENDS TO ZERO

The non-negative derivative implies that increasing $\lambda$ will increase the loss:

$$\frac{\partial \text{Adjusted MSE}}{\partial \lambda} \geq 0.$$

During optimization, the algorithm seeks to minimize the loss, leading to a reduction in $\lambda$. Consequently, $\lambda$ is pushed towards zero. As $\lambda$ approaches zero, $\alpha$ also approaches zero, effectively diminishing the adjusted low-frequency error components.

This behavior contrasts with the GFL approach because:

- **Lack of Separate Adjustments**: By adjusting the combined error rather than the individual outputs, the model cannot balance the adjustments between $y^{\text{train}}$ and $y$.

- **Unidirectional Influence**: The derivative being non-negative means that the optimization consistently pushes $\lambda$ downward without reaching an optimal balancing point.

- **Over-suppression of Low-Frequency Errors**: As $\lambda$ decreases, low-frequency errors are suppressed, potentially ignoring important discrepancies in the low-frequency components.

Therefore, directly applying $\lambda$ to the MSE in the frequency domain results in $\lambda$ tending to zero, leading to suboptimal adjustments of the frequency components.

# B  EXPERIMENTAL ANALYSIS OF GRAPH FOURIER LOSS

In Cylinderflow, we conducted additional analyses to examine the effects of Graph Fourier Loss (GFL).

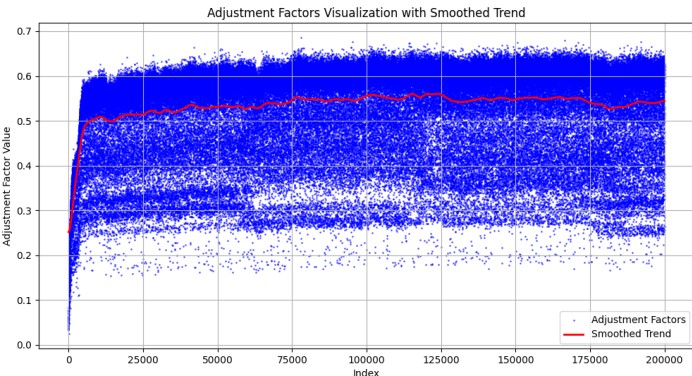

Figure 4: Adjustment Factors Visualization with Smoothed Trend. The scatter plot shows the adjustment factors over time, with a smoothed trend line in red demonstrating the convergence towards 0.5.

We visualized the adjustment factors, as shown in Figure 4. The adjustment factors rapidly converged to approximately 0.5, indicating that GFL reduces the emphasis on low-energy components in the loss function. This mechanism allows the model to prioritize high-energy components during optimization, improving both the overall signal quality and the model's robustness.

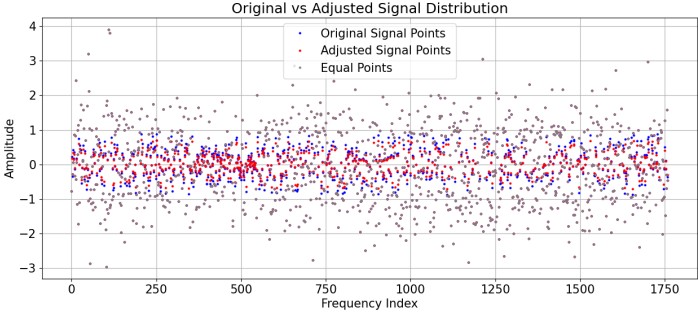

Figure 5: Original vs Adjusted Signal Distribution. The blue points represent the original signal values, and the red points represent the signal values after adjustment by GFL. Equal points are shown in gray, emphasizing areas where the original and adjusted signals coincide.

Additionally, we visualized the frequency domain information post-GFL application in Figure 5. The x-axis represents frequency (low to high), and the y-axis represents energy magnitude. The adjustment factors scale the low-energy components, while the high-energy components remain unaffected. The visualization demonstrates that the adjusted signals (depicted by red dots) exhibit significant energy alterations in the low-energy region, resulting in a smoother and more concentrated signal performance in the frequency domain.

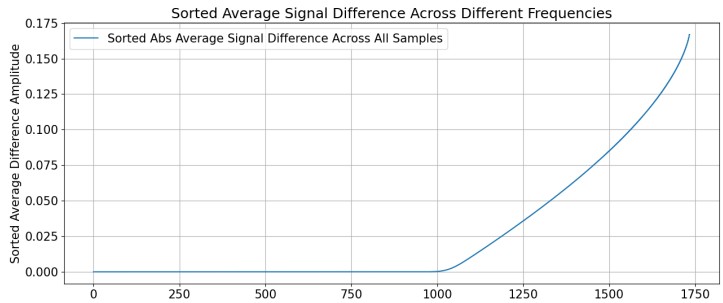

Figure 6: Sorted Average Signal Difference Across Different Frequencies. The plot shows the sorted absolute average signal difference across all samples, highlighting how GFL impacts frequency domain signals.

To quantify the impact of this adjustment, we calculated the average difference in frequency domain information before and after applying GFL, over 100,000 steps following model convergence. The results, presented in Figure 6, show an average difference of 0.067 compared to an original signal mean of 0.72. This indicates that GFL significantly alters the signal representation. The first half of the figure illustrates a straight line, corresponding to the unaltered high-energy components.

| Edge Removal (%) | High Energy Mean | Low Energy Mean | Energy Ratio |
|---|---|---|---|
| 1 | 3.3886 ± 0.0147 | 3.3913 ± 0.0150 | 1.0008 ± 0.0004 |
| 5 | 3.3876 ± 0.0142 | 3.3897 ± 0.0145 | 1.0006 ± 0.0004 |
| 10 | 3.3872 ± 0.0140 | 3.3894 ± 0.0143 | 1.0007 ± 0.0004 |

Table 4: Energy distributions under varying graph connectivity.

(Reviewer ZnqH) To evaluate GFL's robustness under varying graph connectivity, we simulate edge perturbations and measure energy distribution (Table 4). GFL operates on energy distributions rather than precise eigenvectors, ensuring robustness to small connectivity changes. The stable energy ratio confirms its generalization across dynamic graph structures.

## C   THEORETICAL ANALYSIS OF GRAPH FOURIER LOSS

The GFL is defined as:

$$\text{GFL}(\theta) = \frac{1}{N} \left\| \text{adjust}(U^\top y^{\text{train}}(\theta), \alpha) - \text{adjust}(U^\top y, \alpha) \right\|_2^2 = \frac{1}{N} \|e\|_2^2 = \frac{1}{N} \sum_{i=1}^{N} e_i^2.$$

Use the Chain Rule:

$$\frac{\partial \text{GFL}}{\partial \theta} = \frac{2}{N} \sum_i e_i \cdot \frac{\partial e_i}{\partial \theta}.$$

Calculate $\frac{\partial e}{\partial \theta}$: Based on different cases:

Case 1 ($i \in L_{\hat{y}} \cap L_y$):

$$e_i = \alpha(\hat{y}_i - y_i) \Rightarrow \frac{\partial e_i}{\partial \theta} = \alpha \frac{\partial \hat{y}_i}{\partial \theta}.$$

Case 2 ($i \in L_{\hat{y}} \cap H_y$):

$$e_i = \alpha \hat{y}_i - y_i \Rightarrow \frac{\partial e_i}{\partial \theta} = \alpha \frac{\partial \hat{y}_i}{\partial \theta}.$$

Case 3 ($i \in H_{\hat{y}} \cap L_y$):

$$e_i = \hat{y}_i - \alpha y_i \Rightarrow \frac{\partial e_i}{\partial \theta} = \frac{\partial \hat{y}_i}{\partial \theta} - y_i \frac{\partial \alpha}{\partial \theta}.$$

Case 4 ($i \in H_{\hat{y}} \cap H_y$):

$$e_i = \hat{y}_i - y_i \Rightarrow \frac{\partial e_i}{\partial \theta} = \frac{\partial \hat{y}_i}{\partial \theta}.$$

By synthesizing the above steps, we have:

$$\frac{\partial \text{GFL}}{\partial \theta} = \frac{2}{N} \sum_i e_i \cdot \frac{\partial e_i}{\partial \theta} = \frac{2}{N} \left( \sum_{i \in L_{\hat{y}} \cap L_y} \alpha e_i \frac{\partial \hat{y}_i}{\partial \theta} + \sum_{i \in L_{\hat{y}} \cap H_y} \alpha e_i \frac{\partial \hat{y}_i}{\partial \theta} + \sum_{i \in H_{\hat{y}} \cap L_y} (e_i - \alpha y_i) \frac{\partial \hat{y}_i}{\partial \theta} \right).$$

## C.1 RESULT ANALYSIS

- **Weight Adjustment**: The adjustment factor $\alpha$ serves as a frequency-specific weight, modulating the contribution of each frequency component based on its relative importance.

- **Frequency Domain Learning**: By including $\frac{\partial \hat{y}_i}{\partial \theta}$, the gradient integrates frequency domain information, allowing the model to better capture meaningful patterns across different frequencies.

- **Adaptive Learning**: The interaction between $\alpha$ and the error terms ensures that the learning process adaptively shifts focus towards the most relevant frequency components for the task at hand.

## D DATASET DETAILS

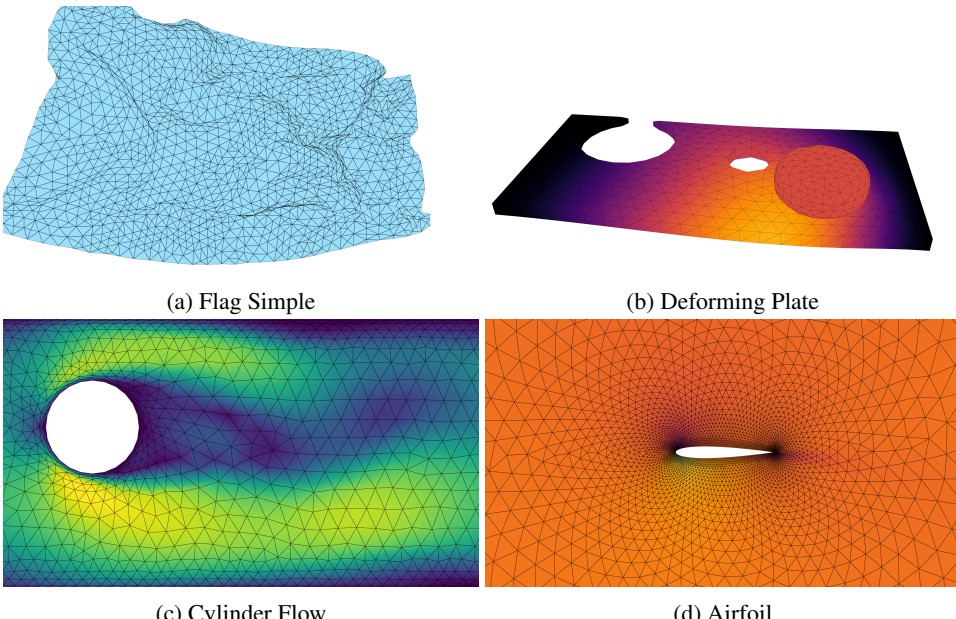

(a) Flag Simple                    (b) Deforming Plate

(c) Cylinder Flow                  (d) Airfoil

Figure 7: Visualization of different datasets.

Table 5: Dataset Specifications

| Dataset | System | Mesh Type | Dimensions | # Steps | time step $\Delta t$ |
|---|---|---|---|---|---|
| FlagSimple | Lagrangian | triangle | 3D | 400 | 0.02 |
| DeformingPlate | Lagrangian | tetrahedral | 3D | 400 | — |
| CylinderFlow | Eulerian | triangle | 2D | 600 | 0.01 |
| Airfoil | Eulerian | triangle | 2D | 600 | 0.008 |

**Dataset Specifications**   Our models are trained and evaluated across four distinct datasets: *FlagSimple*, *DeformingPlate*, *CylinderFlow*, and *Airfoil*. Each dataset consists of 1000 training trajectories, 100 validation trajectories, and 100 test trajectories, with each trajectory comprising between 250 to 600 time steps. The *FlagSimple* dataset models a flag fluttering in the wind using a static Lagrangian mesh with a fixed topology, ignoring collision effects. The *DeformingPlate* dataset simulates the deformation of a hyper-elastic plate driven by a kinematic actuator with a quasi-static simulator, structured on a Lagrangian tetrahedral mesh. The *CylinderFlow* dataset involves the simulation of incompressible The *Airfoil* dataset focuses on the aerodynamic properties around an airfoil section, utilizing a 2D Eulerian mesh to track changes in momentum and density over time. Table 5 details for each dataset include: **System**, indicating whether the simulation is Lagrangian for solid mechanics or Eulerian for fluid dynamics; **Mesh Type**, specifying the geometric configuration such as triangular or tetrahedral; **Dimensions**, indicating whether the simulation is in 2D or 3D; and **# Steps**, the total number of simulation steps in each trajectory reflecting the depth of time-dependent analysis. The **time step** $\Delta t$ column specifies the simulation time increment between each step.

# E   MODEL DETAILS

**Model Hyperparameters**   We employ a batch size of 1 but gradient accumulation of 20 for training and set the Message Passing (MP) time to 15 steps. These configurations are directly adapted from the MGN model, as our model is a further improvement based on MGN. Unlike MGN, which was trained for 10 million steps, we found that our model converged with the above settings in just 5 million steps, allowing us to reduce the training duration. The Adam optimizer is used with an initial learning rate of $10^{-4}$, which decays exponentially to $10^{-6}$ over the course of 2 million training steps, out of a total of 5 million steps. The model comprises four functions: $f_1$, $f_2$, $f_3$, and $f_5$, each configured as a ReLU-activated two-hidden-layer MLP. All the layers are sized at 128, the same as other baselines. The Historical Message-Passing Attention mechanism implemented uses four heads and includes a dropout rate of 0.1. The segmentation rate, denoted by $s_r$, is set at 0.5. Rather than employing a manual setting of the parameter $\lambda$, we have chosen to utilize learnable lambda parameters. These settings are consistently applied across all datasets. Other models utilize the default configurations from their respective papers.

Table 6: Model Input and Output Specifications

| Dataset | edge inputs $e_{ij}^{M}$ | edge inputs $e_{ij}^{W}$ | node inputs $v_i$ | output |
|---|---|---|---|---|
| FlagSimple | $x_{m,ij}, |x_{m,ij}|, x_{w,ij}, |x_{w,ij}|$ | $x_{w,ij}, |x_{w,ij}|$ | $n_i, \dot{x}_i$ | $\ddot{x}_i$ |
| DeformingPlate | $x_{m,ij}, |x_{m,ij}|, x_{w,ij}, |x_{w,ij}|$ | $x_{w,ij}, |x_{w,ij}|$ | $n_i$ | $\dot{x}_i, \sigma_i$ |
| CylinderFlow | $x_{w,ij}, |x_{w,ij}|$ | – | $n_i, w_i$ | $\dot{w}_i$ |
| Airfoil | $x_{w,ij}, |x_{w,ij}|$ | – | $n_i, w_i, \rho_i$ | $\dot{w}_i, \dot{\sigma}_i$ |

**Model Input and Output**   In table 6, several specific terms and symbols define the structure of input and output data for each dataset involved in the simulations. The edge inputs $e_{ij}^{M}$ and $e_{ij}^{W}$ represent interactions associated with edges between nodes $i$ and $j$, where $x_{m,ij}$ denotes the world edge position and $x_{w,ij}$ indicates the mesh edge position. The node inputs $v_i$ include $n_i$, representing node types, and $x_i$, indicating node positions. Other node-specific properties include momentum ($w_i$) and density ($\rho_i$), while outputs encompass acceleration ($\ddot{x}_i$), velocity ($\dot{x}_i$), and von Mises stress ($\sigma_i$).

Table 7: Noise Scale and World Edge Radius Specifications

| Dataset | Noise Scale | World Edge Radius $r_w$ |
|---|---|---|
| FlagSimple | pos: 1e-3 | — |
| DeformingPlate | pos: 3e-3 | 0.03 |
| CylinderFlow | momentum: 2e-2 | — |
| Airfoil | momentum: 1e1, density: 1e-2 | — |

**Noise Injection and World Edge Radius Settings**    To enhance the robustness of our model against noisy inputs and to simulate real-world data conditions more accurately, we implemented a strategy for noise injection into the training process. These noise scales are consistent with the settings used in MeshGraphNet (5), as detailed in Table 7. Additionally, the **world edge radius** $r_w$ column specifies the radius used for defining mesh interactions in the DeformingPlate dataset. (Reviewer YBDW) GFL uses the original topology from the dataset so $r_w$ do not participate in the Laplacian eigendecomposition. This ensures that the precomputation remains efficient and does not incur additional unnecessary overhead.

## F    MEMORY CONSUMPTION AND COMPUTATIONAL SPEED

| Measurements | MGN (5) | HMIT (ours) | optimized HMIT (ours) |
|---|---|---|---|
| $t_{\text{train}}/\text{step}\,[\text{ms}]$ | 4.59 | 8.12 | 5.24 |
| $t_{\text{infer}}/\text{step}\,[\text{ms}]$ | 1.91 | 3.37 | 1.95 |
| Train RAM [GB] | 1.50 | 4.45 | 1.92 |
| Infer RAM [GB] | 0.61 | 0.79 | 0.64 |

Table 8: Comparative study on the CylinderFlow dataset, evaluating computational efficiency and memory usage across methods. (Reviewer ZnqH) The integration of KV cache and dynamic weighted value selection significantly improves computational speed and reduces memory consumption.

Despite the promising advancements offered by the Historical Message-Passing Integration Transformer (HMIT) in simulating physical systems, its initial implementation faced notable limitations in computational speed and memory consumption. To address these challenges, we incorporated two key optimizations. First, **KV Cache** eliminates redundant computations by caching key and value matrices, reducing the attention complexity from $O(3Nsd^2 + 3Nsd)$ to $O(Nsd^2 + 3Nsd)$. Second, **Dynamic Weighted Value Selection** dynamically selects the first $m$ rows of the weighted value matrix $w \in \mathbb{R}^{s \times d}$, where $m$ corresponds to the current message-passing step, further enhancing computational efficiency. As shown in Table 8, these optimizations significantly reduce training and inference times while decreasing memory requirements.

## G    PERFORMANCE OF HMIT ON DAM FLOW

| Measurements | HMIT (ours) | MGN (5) | BSMS (50) | TIE (56) |
|---|---|---|---|---|
| RMSE-1 | **1.08 E-01** | 2.25 E-01 | 1.87 E-01 | 1.62 E-01 |
| RMSE-50 | **2.57 E-01** | 6.03 E-01 | 5.34 E-01 | 5.71 E-01 |
| RMSE-all | **4.63 E-01** | 9.25 E-01 | 8.36 E-01 | 8.49 E-01 |

Table 9: Comparison of HMIT with MGN, BSMS, and TIE on the Dam Flow dataset. HMIT demonstrates superior performance across all metrics.

(Reviewer L6SR) We additionally conducted an experiment on the Dam Problem from the CFD-Bench (68) benchmark, which models the rapid release of water from a column collapse and represents complex free-surface flows with varying velocities. Our method outperforms previous methods (MGN, BSMS, and TIE) across all three RMSE metrics—RMSE-1, RMSE-50, and RMSE-all—by 33% to 57%.

# H  ROLLOUT VISUALIZATIONS

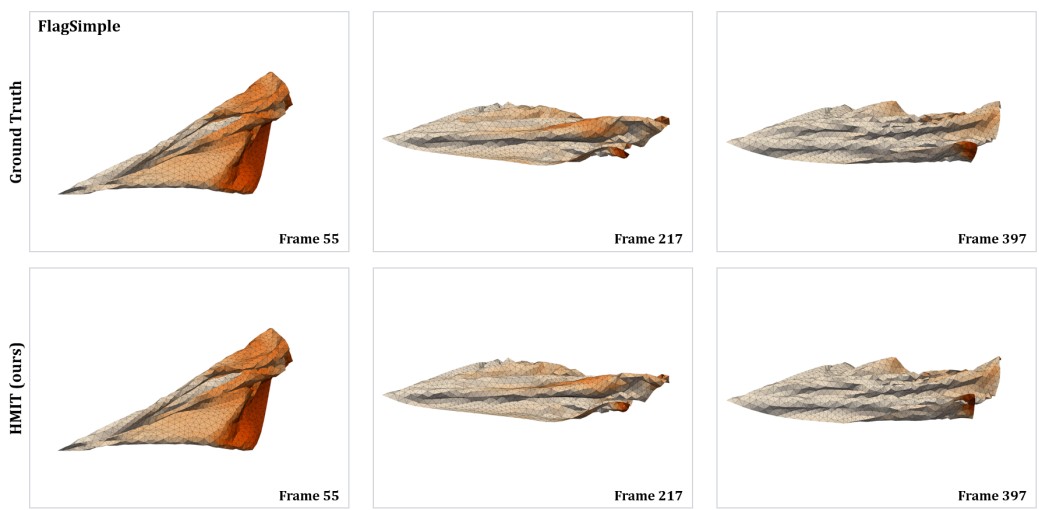

Figure 8: Flag Simple Visualization

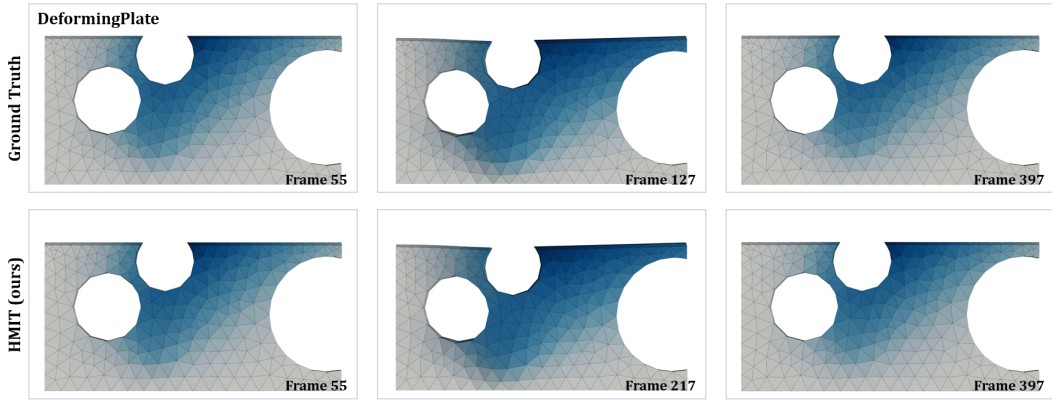

Figure 9: Plate Visualization

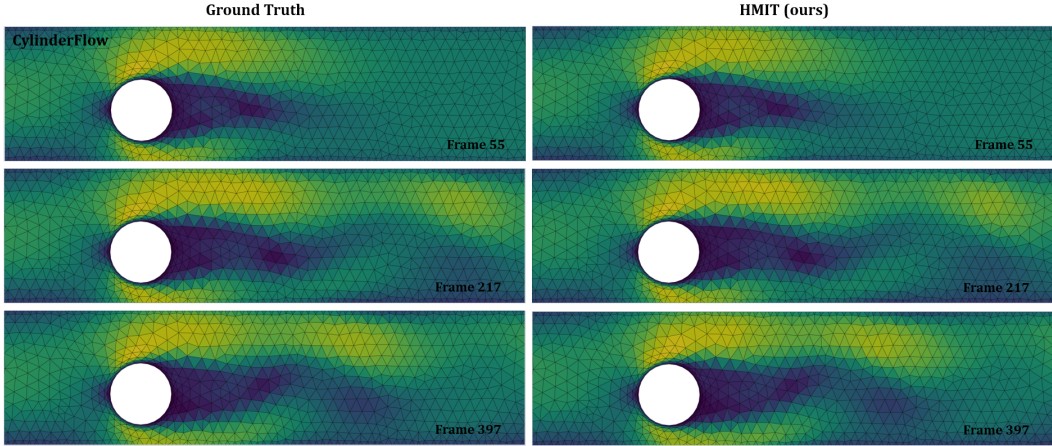

Figure 10: Cylinder Visualization

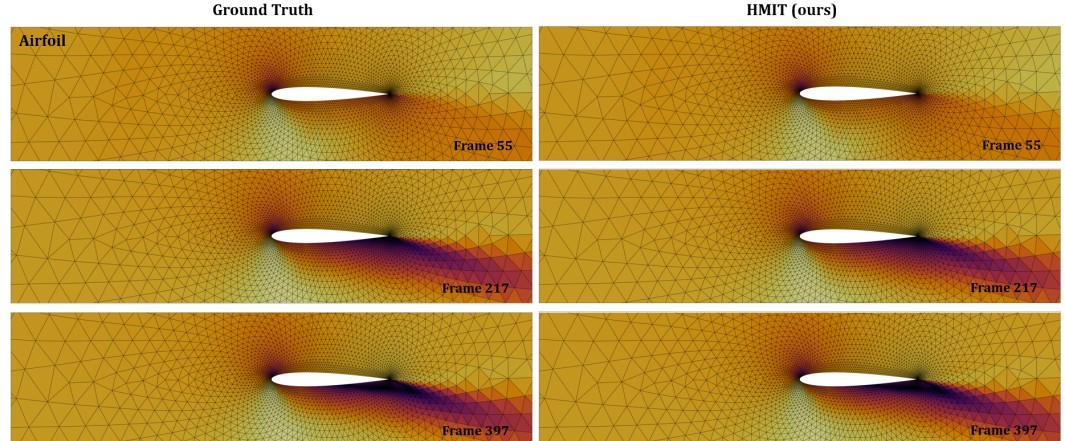

Figure 11: Airfoil Visualization

