# OpenReview forum: "Learning Physical Simulation with Message Passing Transformer"
_ICLR.cc/2025/Conference — Submitted to ICLR 2025_

### Official Review · Reviewer_ZnqH · 2024-10-28

**Soundness:** 3
**Presentation:** 2
**Contribution:** 2
**Rating:** 6
**Confidence:** 3

**Summary:**

The paper applies Hadamard-Product Attention and a Graph Fourier Loss within a ‘Message Passing Transformer’ framework to address feature importance in both the spatial and spectral domains. The authors show that this leads to accuracy improvements for modelling dynamical systems.

**Strengths:**

- The Graph Fourier Loss is a novel loss term for learning dynamic systems (to the best of my knowledge). It is well motivated and discussed in detail throughout the paper and in the appendix. Additionally, it is shown to improve performance on a range of datasets over some baselines.

- The ablation studies clearly show the performance improvements of incorporating the attention mechanism and GFL, highlighting the benefits of the outlined contributions.

**Weaknesses:**

- The main weakness in the paper is the impractical computation runtime of the approach which is substantially slower than the baselines (Table 6). Given that methods such as MGN [2] also focus on achieving substantial computational speed ups, it is not clear from the paper what the practical benefit is in improving the accuracy over such a method whilst being much slower and having larger memory requirements. Additionally, the paper hides some of the training time in a pre-processing step calculating the eigenvalues and eigenvectors. In order to have a fair comparison, the paper should quote this pre-processing time as its a required step for training.

- Using the Hadamard-Product Attention has already been shown to be beneficial in graph representation learning over dot-product attention (Appendix F of [5]). Additionally, the use of message-passing and a transformer itself is not new. For example, GraphGPS [3] uses both local message-passing and global transformer updates. Graph Attention Networks [4] uses attention to weight local updates. Graph MLP-MIXER [5] uses Hadamard-Product Attention between local patch encodings. The paper needs to be more clear what are its contributions in relation to other Graph Neural Networks (such as those which I have listed and others). It seems the only difference and contribution is the Graph Fourier Loss which can lead to a computational burden in the pre-processing step and suffers from instabilties (see Question 1).

**Questions:**

-  Small perturbations to the Laplacian could result in completely different eigenspaces [1]. Do you think the introduction then of a Graph Fourier loss (GFL) would lead to difficulties for remeshing or to generalise when the underlying connectivity/mesh differs?

- There is quite substantial differences between the results of MGN in your work and that quoted in the original paper [2] on the same datasets. Why is this the case?

- When you do ablations and state your results "without" GFL it is not clear what you mean. Do you mean replacing this loss with an MSE on the velocities?

- You state in the introduction that a common solution to over-smoothing and over-squashing in similar works is to map to a latent space before preprocessing. However, if you map node features to a latent representation and then use a standard GNN which only incorporates local updates then this would not solve over-squashing. How does your method solve this issue?

- The words "new" and "universal" to describe your architecture are not clear to me. Is your model universal? Additionally, the use of message-passing and a transformer itself is not new (see second point of weaknesses). What is novel about the architecture which you use outside of the loss function?

- Given the above point and the many other methods which use attention and message-passing, the term 'message passing transformer' to describe the method is not clear and confuses other terms and methods used in the Graph Machine Learning literature. Why would you say the term 'message passing transformer' describes your architecture better than what is done with these other approaches?

- You mention in the paper that HPA and GFL "synergistically" optimize the model’s performance. Why is it the case the the methods in combination are more effective than the sum of their parts?

- Some recent work [6] has also shown improvements over these baselines on these datasets. Can you say that your model is 'State-of-the-art' without also having newer methods such as this as a baseline?

[1] Huang et al. ON THE STABILITY OF EXPRESSIVE POSITIONAL ENCODINGS FOR GRAPHS. ICLR 2024.

[2] Pfaff et al. LEARNING MESH-BASED SIMULATION WITH GRAPH NETWORKS. ICLR 2021.

[3] Rampášek et al. Recipe for a General, Powerful, Scalable Graph Transformer. NeurIPS 2022.

[4] Veličković et al. Graph Attention Networks. ICLR 2018.

[5] He et al. A GENERALIZATION OF VIT/MLP-MIXER TO GRAPHS. ICML 2023.

[6] Luo et al. CARE: Modeling Interacting Dynamics Under Temporal Environmental Variation. NeurIPS 2023.

---

> ### Author Response · Authors · 2024-11-21
> **Individual response Part 1.**
>
> Thank you for your insightful comments and constructive feedback. Below, we provide detailed responses to each point raised:
>
>
> **1. Quote Graph Laplacian pre-processing time**
> **Response**:
> **[Experiment: Precomputation Cost]**:
>
> | Dataset   | Eigen Decomposition Time per Sample| Total Processing Time  |
> |-----------|------------------------------------|------------------------|
> | Cylinder  | 0.026 s                            | 27.21 s                |
> | Airfoil   | 0.204 s                            | 218.34 s               |
> | Plate     | 0.016 s                            | 19.19 s                |
> | Flag      | 0.020 s                            | 21.82 s                |
>
> While we provide a precomputation cost table for clarity, this data processing only occurs once and can be considered part of the dataset generation process.
>
> The primary computational cost is in the eigen decomposition of the graph Laplacian, with a total precomputation time of about 3 minutes—negligible compared to training time. For the four datasets with static graph topologies, eigen decomposition is only needed at $t=0$ rather than at each time step.
>
> For datasets with dynamic topologies, eigen decomposition is required at each time step but only needs to be done once, in batch, before training, and cached for reuse. This approach reduces on-the-fly calculation complexity, improving training efficiency. Importantly, inference does not require precomputed eigenvalues or eigenvectors, so inference speed remains unaffected.
>
>
> **2. Small Laplacian perturbations could lead to distinct eigenspaces. Could GFL hinder remeshing/generalization with varying graph connectivity?**
> **Response**: Thank you for pointing this out. Our GFL remains robust under varying graph connectivity due to the  Energy-Based Robustness: GFL operates on frequency-domain energy distributions ($E$) rather than relying directly on eigenvector precision. By focusing on mean energy across high- and low-energy components, it reduces sensitivity to small eigenvector perturbations caused by narrow eigenvalue gaps.
>
> **[Experiment: GFL Robustness Performance]**:
>
> | Edge Removal Percentage  | High Energy Mean          | Low Energy Mean           | Energy Ratio (High/Low)        |
> |--------------------------|---------------------------|---------------------------|--------------------------------|
> | 1%                       | 3.388615 ± 0.014727       | 3.391265 ± 0.014988       | 1.000784 ± 0.000418            |
> | 5%                       | 3.387550 ± 0.014215       | 3.389730 ± 0.014485       | 1.000643 ± 0.000392            |
> | 10%                      | 3.387155 ± 0.013993       | 3.389425 ± 0.014275       | 1.000670 ± 0.000385            |
>
>
> **3. Notable discrepancies exist between MGN results in your work and those in the original paper on the same datasets. Why?**
> **Response**:
> We utilized the [`mgn_pytorch`](https://github.com/echowve/meshGraphNets_pytorch) implementation to run MGN baseline, yielding the results shown in the table.
> There may be differences between our results and those in the original paper, possibly due to some variations between PyTorch and TensorFlow. Notice that we use RMSE(1E-2) to visualize and MGN use RMSE x$10^{-3}$, and FlagSimple's results seems substantially different because we used speed, whereas MGN used position. We note that other papers using MGN as a baseline[1][2][7] also show numbers that are quite substantially different.
>
> | Measurements    | Dataset          | Original MGN Result | Our's MGN Result| BSMS’s MGN[7] | Mesh Transformer’s MGN[1]  | CARE’s MGN[2] |
> |-----------------|------------------|---------------------|-----------------|------------|-------------------------|------------|
> | **RMSE-1**      | cylinder         | 0.00234             | 0.00524         | 0.00226    | 0.0058                  | 0.1434     |
> |                 | airfoil          | 0.314               | 3.14            | 0.435      | -                       | 0.0385     |
> |                 | plate            | 0.00025             | 0.000269        | 0.000198   | -                       | -          |
> | **RMSE-50**     | cylinder         | 0.0063              | 0.014           | 0.0439     | 0.0405                  | 0.4328     |
> |                 | airfoil          | 0.582               | 5.36            | 16.6       | -                       | 0.0751     |
> |                 | plate            | 0.0018              | 0.00173         | 0.000288   | -                       | -          |
> | **RMSE-all**    | cylinder         | 0.04088             | 0.0432          | 0.107      | -                       | -          |
> |                 | airfoil          | 11.529              | 20.8            | 69.6       | -                       | -          |
> |                 | plate            | 0.0151              | 0.0161          | 0.00151    | -                       | -          |

---

> > ### Author Response · Authors · 2024-11-21
> > **Individual response Part 2.**
> >
> > **4. Your ablation studies use the term "without" GFL, but what does that mean? Are you using MSE on velocities instead?**
> > **Response**: When GFL is not used, we compute MSE on the model's entire output. We have updated this point in the paper.
> >
> > **5.You mention latent space mapping helps address over-squashing, but latent node feature mapping alone wouldn’t resolve this issue in GNNs. How does your method manage this?**
> > **Response**: Our approach mitigates over-squashing by using HPA to extend each node’s accessible information across multiple MP steps, allowing multi-step aggregation for broader context. This integration of information from earlier steps enables the model to more clearly capture propagation over varying distances, effectively propagating essential node features throughout the network. We will revise the paper's Introduction and Method sections to more clearly position our approach.
> >
> > **6. Terms like "new" and "universal" to describe your model’s architecture are unclear. What is truly novel about your approach aside from GFL?**
> > **Response**:
> > * We acknowledge that describing our model as "universal" was imprecise. Our approach is specifically tailored for mesh-based structural representations, making it versatile within this domain but not generalizable to all graph types or non-mesh structures.
> > * Our approach introduces HPA by integrating multi-step information from historical message passing, applying Hadamard products with feature-dimensional softmax, and leveraging a decoder-only architecture. This design mitigates over-squashing, captures fine-grained information, and achieves linear scalability with the number of nodes.
> >
> > We revised the paper's Abstract, Introduction and Method sections to more clearly position our approach.
> >
> > **7. The term "message passing transformer" doesn’t clearly set your model apart. Why do you think this term suits your method?**
> > **Response**:Thank you for your valuable feedback. We’ve renamed Message Passing Transformer to **Historical Message-Passing Integration Transformer (HMIT)** and Hadamard-Product Attention to **Historical Message-Passing Attention (HMPA)** to emphasize its unique integration of historical messages, which stands out by:
> > 1. Integrating historical message passing information to mitigate over-squashing.
> > 2. Using feature-dimensional softmax with Hadamard products for fine-grained information capture.
> > 3. Adopting a decoder-only design for linear scalability with the number of nodes.
> > 4. Applying Graph Fourier Loss (GFL) for frequency-based information modulation.
> >
> > We have updated the **Title** to "Learning Physical Simulation with Historical Message-Passing Integration Transformer" and revised the Abstract, Introduction, and Method sections to emphasize the specific techniques we use and the problems they address.
> >
> > **8. You mention that HPA and GFL "synergistically" optimize model performance. Why is the combination of these methods more effective than each alone?**
> > **Response**:
> > * Our ablation results confirm that combining HPA and GFL yields optimal performance by addressing complementary aspects of the learning process.
> > * HPA enhances spatial representation, allowing fine-grained, feature-level adjustments during message passing, which improves node embeddings by dynamically focusing on relevant information.
> > * GFL operates in the spectral domain, balancing frequency components to capture essential structural patterns.
> > * Together, HPA and GFL ensure critical information is captured across both spatial and spectral domains, leading to a more robust model than either method alone.
> >
> > We have clarified the synergy and also replaced "synergistically" with "jointly" in the paper to make the language less exaggerated.

---

> > > ### Author Response · Authors · 2024-11-21
> > > **Individual response Part 3.**
> > >
> > > **9. Lack of comparison against newer methods?**
> > > **Response**: We have compared our approach to state-of-the-art message-passing GNNs(MGN 2021, BSMS 2023) and Transformer-based method(TIE 2022). However, to enable a more comprehensive comparison, we have added Mesh-Transformer[1] and Graph MLP-MIXER[6] as additional baselines for testing. We additionally test HCMT[8] only on DeformingPlate dataset because its CMT and HMT modules were ultimately designed specifically for collision cases. Papers[3][4][5][6] you provided are not in the GNN4Sim domain, so we only select [6] as an additional baseline. However, we have already cited these four papers and [2] as highly valuable previous works.
> > >
> > >  **[Experiment: Additional Baslines Comparison]**:
> > >
> > > | Measurements    | Dataset   | Ours (MPT)         |Mesh Transformer[1]| HCMT[2]          |Graph MLP-MIXER[3]|
> > > |-----------------|-----------|--------------------|-------------------|------------------|------------------|
> > > | RMSE-1 [1E-2]   | Cylinder  | **2.03 E-01**      | 3.05 E-01         | -                | 4.12 E-01        |
> > > |                 | Airfoil   | **2.61 E+02**      | 2.96 E+02         | -                | 3.05 E+02        |
> > > |                 | Plate     | **1.00 E-02**      | 2.38 E-02         | 1.15 E-02        | 3.28 E-02        |
> > > |                 | Flag      | **1.12 E-02**      | 4.73 E-02         | -                | 6.89 E-02        |
> > > | RMSE-50 [1E-2]  | Cylinder  | **6.32 E-01**      | 1.07              | -                | 4.61             |
> > > |                 | Airfoil   | **4.08 E+02**      | 5.21 E+02         | -                | 6.32 E+02        |
> > > |                 | Plate     | **9.25 E-02**      | 1.30 E-01         | 9.68 E-01        | 6.19 E-01        |
> > > |                 | Flag      | **1.87**           | 2.04              | -                | 3.90             |
> > > | RMSE-all [1E-2] | Cylinder  | **3.78**           | 4.26              | -                | 2.05 E+01        |
> > > |                 | Airfoil   | **1.64 E+03**      | 2.00 E+03         | -                | 3.97 E+03        |
> > > |                 | Plate     | **1.09**           | 1.28              | 1.15             | 8.22             |
> > > |                 | Flag      | **2.05**           | 2.32              | -                | 7.25             |
> > >
> > > [1] Eagle: Large-scale learning of turbulent fluid dynamics with mesh transformers
> > > [2] CARE: Modeling Interacting Dynamics Under Temporal Environmental Variation
> > > [3] ON THE STABILITY OF EXPRESSIVE POSITIONAL ENCODINGS FOR GRAPHS
> > > [4] Recipe for a General, Powerful, Scalable Graph Transformer
> > > [5] Graph Attention Networks
> > > [6] A GENERALIZATION OF VIT/MLP-MIXER TO GRAPHS
> > > [7] Efficient Learning of Mesh-Based Physical Simulation with Bi-Stride Multi-Scale Graph Neural Network
> > > [8] Learning flexible body collision dynamics with hierarchical contact mesh transformer

---

> > > > ### Comment · Reviewer_ZnqH · 2024-11-23
> > > > **Response to Rebuttal by Authors**
> > > >
> > > > Thank you very much for your detailed responses to my questions! The additional baselines run are considerable and very much appreciated and I think the changes you have suggested would be beneficial. Are the authors planning on uploading the revised manuscript for us to see these changes? Additionally, I still have a couple of outstanding clarifications based on my initial review - from the two points in the weaknesses section. I know these aren't questions but given these are my main barriers to accepting the paper from the initial review, it would be greatly appreciated to hear your thoughts with regards to these points and whether you think they are fair.
> > > >
> > > > -  _"Given that methods such as MGN [2] also focus on achieving substantial computational speed ups, it is not clear from the paper what the practical benefit is in improving the accuracy over such a method whilst being much slower and having larger memory requirements."_
> > > >
> > > > - _"Using the Hadamard-Product Attention has already been shown to be beneficial in graph representation learning over dot-product attention (Appendix F of [5]). Additionally, the use of message-passing and a transformer itself is not new. For example, GraphGPS [3] uses both local message-passing and global transformer updates. Graph Attention Networks [4] uses attention to weight local updates. Graph MLP-MIXER [5] uses Hadamard-Product Attention between local patch encodings. The paper needs to be more clear what are its contributions in relation to other Graph Neural Networks. "_
> > > >
> > > > For example, on this last point, it is not clear when you say things such as "To mitigate over-squashing, capture fine-grained details, and scale linearly with node count, we purpose HPA". Given that HPA has been used in graph-learning before, precisely because of these properties.
> > > >
> > > >
> > > > [2] Pfaff et al. LEARNING MESH-BASED SIMULATION WITH GRAPH NETWORKS. ICLR 2021.
> > > >
> > > > [3] Rampášek et al. Recipe for a General, Powerful, Scalable Graph Transformer. NeurIPS 2022.
> > > >
> > > > [4] Veličković et al. Graph Attention Networks. ICLR 2018.
> > > >
> > > > [5] He et al. A GENERALIZATION OF VIT/MLP-MIXER TO GRAPHS. ICML 2023.

---

> > > > > ### Author Response · Authors · 2024-11-23
> > > > > **Response to Comment by Reviewer**
> > > > >
> > > > > **1. Methods like MGN [2] focus on significant computational speed-ups. The paper should justify the practical benefits of improved accuracy over such methods, considering the increased computational and memory costs.**
> > > > >
> > > > > Thank you for your comment regarding computational efficiency and memory requirements compared to methods like MGN [2]. We have addressed this by incorporating two key optimizations:
> > > > >
> > > > > 1. **KV Caching**: By caching key and value matrices, we eliminate redundant computations, reducing the attention complexity from $O(3Nsd^2 + 3Nsd)$ to $O(Nsd^2 + 3Nsd)$.
> > > > >
> > > > > 2. **Dynamic Weighted Value Selection**: We dynamically select the first $m$ rows of the weighted value matrix $w \in \mathbb{R}^{s \times d}$, where $m$ corresponds to the current message-passing step, further improving efficiency.
> > > > >
> > > > > | Measurements                                     | **MGN** [MGN] | **Ours (HMIT)** | **Ours (HMIT KV cache)** |
> > > > > |--------------------------------------------------|---------------|-----------------|--------------------------|
> > > > > | $t_\text{train} / \text{step} \, [\text{ms}]$    | 4.59          | 8.12            | 5.24                     |
> > > > > | $t_\text{infer} / \text{step} \, [\text{ms}]$    | 1.91          | 3.37            | 1.95                     |
> > > > > | Train RAM [GB]                                   | 1.50          | 4.45            | 1.92                     |
> > > > > | Infer RAM [GB]                                   | 0.61          | 0.79            | 0.64                     |
> > > > >
> > > > > **Table**: Comparative study on the CylinderFlow dataset, evaluating computational efficiency and memory usage across methods, including the benefits of KV cache.
> > > > >
> > > > > **2. Using Hadamard-Product Attention has been shown to benefit graph representation learning over dot-product attention (Appendix F of [5]). Moreover, combining message-passing and transformers is not new: GraphGPS [3] integrates local message-passing with global transformer updates, GAT [4] uses attention to weight local updates, and Graph MLP-MIXER [5] applies Hadamard-Product Attention to local patch encodings. The paper should clarify its unique contributions relative to these Graph Neural Networks.**
> > > > >
> > > > > We previously referred to our method as HPA, but to avoid potential ambiguity, we have renamed it **Historical Message-Passing Attention (HMPA)**. HMPA integrates **historical message-passing information**, **feature-dimensional softmax**, **Hadamard products**, and **decoder-only attention** to achieve three key objectives: **mitigating over-squashing**, **capturing fine-grained details**, and **scaling linearly with the node count**.
> > > > >
> > > > > Our method differs from the Hadamard Attention in [5] in the following key aspects:
> > > > >
> > > > > 1. **Localized vs. Graph-Wide Operations**: The method in [5] applies an $N \times N$ adjacency matrix $A^P$ to encode graph-wide relationships, performing updates across the entire graph simultaneously. In contrast, our approach focuses on localized computation, constructing sequences from node attributes and aggregated edge features to update individual nodes efficiently.
> > > > >
> > > > > 2. **Flexible Sequence Construction**: Instead of directly relying on $A^P$, we dynamically construct the key and value matrices $K, V \in \mathbb{R}^{s \times d}$ by concatenating node features and aggregated edge attributes across iterations. This design provides flexibility to handle diverse graph structures and edge features.
> > > > >
> > > > > 3. **Feature-Centric Attention**: While [5] applies softmax over the sequence dimension to emphasize positional relationships, our method employs a scaled Hadamard product $v \odot K$ with softmax applied along the feature dimension. This highlights the relative importance of features, enabling more nuanced attention at the feature level.
> > > > >
> > > > > 4. **Scalability and Efficiency**: By avoiding the need for a full $N \times N$ adjacency matrix, our method significantly reduces computational overhead, focusing on node-specific updates. This makes our approach more scalable for large graphs, especially when combined with optimizations like KV caching.
> > > > >
> > > > > Additionally, compared to **GraphGPS**, which uses local message-passing combined with specialized global transformer updates, our method employs local message-passing and local, cross-message-passing decoder-only attention. Furthermore, we incorporate Hadamard products and feature-wise softmax to enhance flexibility and efficiency.
> > > > >
> > > > > Unlike **Graph Attention Networks (GATs)**, where attention is used to assign weights to edges, our method calculates attention weights within the feature space to process historical message-passing information, making it particularly effective for capturing intra-feature relationships across message-passing iterations.
> > > > >
> > > > > [2] Pfaff et al. LEARNING MESH-BASED SIMULATION WITH GRAPH NETWORKS
> > > > >
> > > > > [3] Rampášek et al. Recipe for a General, Powerful, Scalable Graph Transformer
> > > > >
> > > > > [4] Veličković et al. Graph Attention Networks
> > > > >
> > > > > [5] He et al. A GENERALIZATION OF VIT/MLP-MIXER TO GRAPHS

---

> > > > > > ### Comment · Reviewer_ZnqH · 2024-11-23
> > > > > > **Response to further rebuttal**
> > > > > >
> > > > > > Thanks again for the very comprehensive response. I believe the changes and comments have significantly improved the manuscript and I have raised my score accordingly.  I maintain my average confidence score as there are related works which I am not familiar with.

---

> > > > > > > ### Author Response · Authors · 2024-11-27
> > > > > > > **The revised paper has been uploaded**
> > > > > > >
> > > > > > > Thank you for your thoughtful feedback and for raising your score. We would like to inform you that the revised version of the paper has now been uploaded.

---

### Official Review · Reviewer_YBDW · 2024-10-30

**Soundness:** 3
**Presentation:** 3
**Contribution:** 2
**Rating:** 5
**Confidence:** 4

**Summary:**

This paper introduces the Message Passing Transformer for physical simulation. It employs Hadamard-Product Attention for fine-grained feature processing and Graph Fourier Loss for optimization. The model achieves  accuracy gains in long-term simulations of both Lagrangian and Eulerian dynamical systems compared to existing methods.

**Strengths:**

The use of Hadamard-Product Attention (HPA) and Graph Fourier Loss (GFL) enhance the accuracy and robustness of physical simulations. The results show that MPT has superior performance on four datasets.

**Weaknesses:**

1. The paper suggests using Graph Fourier Transform for preprocessing by calculating the eigenvectors of the graph Laplacian matrix, assuming a fixed graph topology. However, for Lagrangian systems like deforming plates or flags, the graph topology changes over time based on world positions and relationships. This inconsistency raises questions about the validity of the methods and experiments presented, as the assumption of a constant structure contradicts the dynamic nature of Lagrangian systems.
2. The introduction of HPA significantly increases computational overhead in training and inference. The computation overhead largely depends on the sequence length, there is a lack of comprehensive computation analysis for Sec 4.2.
3. The proposed method should compare more attention-based baselines, such as [1], [2].
4. There are missing annotations in Figure 3a, making it unclear which one is HPA.

[1] Eagle: Large-scale learning of turbulent fluid dynamics with mesh transformers.
[2] Learning flexible body collision dynamics with hierarchical contact mesh transformer

**Questions:**

1. In line 113, it is said the graph topology does not vary over sequence. However, for lagrangian systems like deforming plate and flag simple that construct lagrangian edges based on world positions under certain radius, how do you compute eigenvectors of the graph laplacian matrix.
2. In more complex scenarios that require mesh refinement, such as the sphere and flag dynamics datasets, the graph topology will also change over time. How can Graph Fourier Loss be applied in these situations?
3. In Sec 4.3, why is the loss calculated on the results after the adjust function rather than directly on the results from the Graph Fourier Transform?

---

> ### Author Response · Authors · 2024-11-21
> **Individual response Part 1.**
>
> Thank you for your insightful comments and constructive feedback. Below, we provide detailed responses to each point raised:
>
> **1. Reviewer: HPA introduces significant computational overhead; a detailed complexity analysis is missing.**
> **Response**:Thank you for pointing this out. We have modified the paper and replaced the section '**Gradient Computation in HPA**' with '**Complexity Analysis**'.
>     **Complexity Analysis**
>         We calculate the FLOPs for the Scaled Hadamard-Product Attention (HPA) mechanism, with $N$ as the number of nodes, $s$ as the sequence length, and $d$ as the feature dimension. Constructing the key and value matrices $K$ and $V$ requires linear transformations across each sequence step, amounting to $2Nsd^2$ FLOPs. The calculation of attention weights $a$, including the element-wise product and softmax operation, requires $2Nsd$ FLOPs. The element-wise multiplication of $a$ and $V$, followed by a linear transformation to reshape the result back to dimension $d$, adds $Nsd + Nsd^2$ FLOPs. Summing these components, the total FLOPs for HPA is $3Nsd^2 + 3Nsd$. This complexity scales **linearly** with the number of nodes $N$, with a fixed sequence length $s$. Compared to a global self-attention mechanism, our decoder-only architecture avoids the $O(N^2)$ complexity, making it especially efficient and advantageous for large-scale simulations.
>
> **2.More attention-based baselines should be compared, such as [1], [2].**
> **Response**:Thank you for your suggestion. Currently, we use state-of-the-art message-passing GNNs(MGN, BSMS) and Transformer-based method(TIE) as our baselines. However, to enable a more comprehensive comparison, we have added Mesh-Transformer[1] and Graph MLP-MIXER[3] as additional baselines for testing. We additionally test HCMT[2] only on DeformingPlate dataset because its CMT and HMT modules were ultimately designed specifically for collision cases. Although MPT is slower, it outperforms all competitors in both short- and long-term predictions.
>
>  **[Experiment: Additional Baslines Comparison]**:
>
> | Measurements    | Dataset   | Ours (MPT)         |Mesh Transformer[1]| HCMT[2]          |Graph MLP-MIXER[3]|
> |-----------------|-----------|--------------------|-------------------|------------------|------------------|
> | RMSE-1 [1E-2]   | Cylinder  | **2.03 E-01**      | 3.05 E-01         | -                | 4.12 E-01        |
> |                 | Airfoil   | **2.61 E+02**      | 2.96 E+02         | -                | 3.05 E+02        |
> |                 | Plate     | **1.00 E-02**      | 2.38 E-02         | 1.15 E-02        | 3.28 E-02        |
> |                 | Flag      | **1.12 E-02**      | 4.73 E-02         | -                | 6.89 E-02        |
> | RMSE-50 [1E-2]  | Cylinder  | **6.32 E-01**      | 1.07              | -                | 4.61             |
> |                 | Airfoil   | **4.08 E+02**      | 5.21 E+02         | -                | 6.32 E+02        |
> |                 | Plate     | **9.25 E-02**      | 1.30 E-01         | 9.68 E-01        | 6.19 E-01        |
> |                 | Flag      | **1.87**           | 2.04              | -                | 3.90             |
> | RMSE-all [1E-2] | Cylinder  | **3.78**           | 4.26              | -                | 2.05 E+01        |
> |                 | Airfoil   | **1.64 E+03**      | 2.00 E+03         | -                | 3.97 E+03        |
> |                 | Plate     | **1.09**           | 1.28              | 1.15             | 8.22             |
> |                 | Flag      | **2.05**           | 2.32              | -                | 7.25             |
>
> **3. Figure 3a annotations are unclear; which line represents HPA?**
> **Response**: Thank you for pointing this out. We apologize for the annotation mistake; the gray line represents Dot-Product Attention, while the red line represents our Hadamard-Product Attention. We have corrected this in the paper accordingly.

---

> > ### Author Response · Authors · 2024-11-21
> > **Individual response Part 2.**
> >
> > **4. Line 113 claims constant graph topology over time, yet in Lagrangian systems (e.g., deforming plate), edges are dynamically constructed. How are Laplacian eigenvectors computed?**
> > **Response**: Thank you for pointing this out. We clarify that our GFL **can** be applied to cases with changing graph topology, as long as GFT is performed during preprocessing for each time step. Relevant sections of the paper have been revised accordingly.
> > **[Experiment: Precomputation Cost]**:
> >
> > | Dataset   | Eigen Decomposition Time per Sample| Total Processing Time  |
> > |-----------|------------------------------------|------------------------|
> > | Cylinder  | 0.026 s                            | 27.21 s                |
> > | Airfoil   | 0.204 s                            | 218.34 s               |
> > | Plate     | 0.016 s                            | 19.19 s                |
> > | Flag      | 0.020 s                            | 21.82 s                |
> >
> > While we provide a precomputation cost table for clarity, this data processing only occurs once and can be considered part of the dataset generation process.
> >
> > The primary computational cost is in the eigen decomposition of the graph Laplacian, with a total precomputation time of about 3 minutes—negligible compared to training time. For the four datasets with static graph topologies, eigen decomposition is only needed at $t=0$ rather than at each time step.
> >
> > For datasets with dynamic topologies, eigen decomposition is required at each time step but only needs to be done once, in batch, before training, and cached for reuse. This approach reduces on-the-fly calculation complexity, improving training efficiency. Importantly, inference does not require precomputed eigenvalues or eigenvectors, so inference speed remains unaffected.
> >
> > **5. For cases requiring mesh refinement, such as sphere and flag datasets, how does GFL handle time-varying graph topology?**
> > **Response**: Our GFL uses the original topology from the dataset, and any added edges, such as the world edges, **do not participate** in the Laplacian eigendecomposition. This ensures that the precomputation remains efficient and does not incur additional unnecessary overhead. Relevant sections of the paper have been revised accordingly.
> >
> > **6. Why is loss calculated after applying the adjust function in Sec 4.3 rather than directly after the Graph Fourier Transform?**
> > **Response**:
> > * Calculating the MSE after performing a Graph Fourier Transform on the result is mathematically equivalent to computing MSE directly. Applying a piecewise function in the frequency domain can modulate the frequency information across all nodes simultaneously.
> > * The adjustment factor $\alpha$, derived from the ratio of mean energy in high- and low-energy partitions, adaptively balances their contributions to the loss.
> > $$
> > \alpha = \sqrt{\frac{\text{mean}(E_{\text{high}})}{\text{mean}(E_{\text{low}}) + \epsilon}} \cdot \lambda
> > $$
> > * Without the adjustment factor $\lambda$, $\alpha$ would excessively boost low-energy components to match high-energy levels, amplifying noise and reducing performance. Introducing $\lambda$ allows adaptively controlled scaling in low-energy regions.
> >
> >
> > [1] Eagle: Large-scale learning of turbulent fluid dynamics with mesh transformers
> >
> > [2] Learning flexible body collision dynamics with hierarchical contact mesh transformer
> >
> > [3] A GENERALIZATION OF VIT/MLP-MIXER TO GRAPHS

---

> > > ### Author Response · Authors · 2024-11-25
> > > **Reminder: Public Discussion Period for Submission 2831**
> > >
> > > Dear reviewer, this is a friendly reminder that the public discussion period for Submission 2831 is ending soon. If you have any questions or concerns about our rebuttal, we’d be happy to address them.  Thank you for your time and feedback!

---

> > > ### Comment · Reviewer_YBDW · 2024-11-26
> > >
> > > I thank the authors for their extensive rebuttal. I think most of my concerns have been addressed. I will raise my score if the authors include the extended results to the revised paper.

---

> > > > ### Author Response · Authors · 2024-11-27
> > > > **The revised paper has been uploaded**
> > > >
> > > > Thank you for your thoughtful feedback and for raising your score. We truly appreciate your time and consideration. We would like to inform you that the revised version of the paper has now been uploaded. For a detailed overview of the changes, you can refer to the latest overall official comment.

---

> ### Author Response · Authors · 2024-11-28
> **Consideration of Further Suggestions for Improving the Score**
>
> Thank you for your thoughtful review and valuable feedback. We appreciate the time you have taken to evaluate our work. We would like to ask if there is anything else we can do to improve the rating of our submission. Your insights would be greatly appreciated as we continue to refine our approach.
>
> We conducted **robustness tests on GFL**, further addressing your concern in Weakness 1 regarding the robustness of GFL:
> Our GFL remains robust under varying graph connectivity due to the  Energy-Based Robustness: GFL operates on frequency-domain energy distributions ($E$) rather than relying directly on eigenvector precision. By focusing on mean energy across high- and low-energy components, it reduces sensitivity to small eigenvector perturbations caused by narrow eigenvalue gaps.
>
> **[Experiment: GFL Robustness Performance on Airfoil]**:
>
> | Edge Removal Percentage  | High Energy Mean          | Low Energy Mean           | Energy Ratio (High/Low)        |
> |--------------------------|---------------------------|---------------------------|--------------------------------|
> | 1%                       | 3.388615 ± 0.014727       | 3.391265 ± 0.014988       | 1.000784 ± 0.000418            |
> | 5%                       | 3.387550 ± 0.014215       | 3.389730 ± 0.014485       | 1.000643 ± 0.000392            |
> | 10%                      | 3.387155 ± 0.013993       | 3.389425 ± 0.014275       | 1.000670 ± 0.000385            |
>
>
> Additionally, in our discussions with Reviewer ZnqH, we further optimized the computational efficiency of our method, significantly reducing the increase in time and memory usage compared to MGN. This means that the wall clock time and memory footprint of our model are **no longer limiting factors you mentioned in Weakness 2**. As shown in the table below, the optimized version of HMIT achieves substantial improvements in both training and inference time, while also reducing memory usage:
>
>
> **[Experiment: Memory Consumption and Computational Speed]**:
> | Measurements                             | MGN | HMIT (ours) | optimized HMIT (ours) |
> |------------------------------------------|-----------|-------------|-----------------------|
> | $t_\text{train} / \text{step} \, [\text{ms}]$ | 4.59      | 8.12        | 5.24                  |
> | $t_\text{infer} / \text{step} \, [\text{ms}]$ | 1.91      | 3.37        | 1.95                  |
> | Train RAM [GB]                           | 1.50      | 4.45        | 1.92                  |
> | Infer RAM [GB]                           | 0.61      | 0.79        | 0.64                  |
>
> Despite the promising advancements offered by the Historical Message-Passing Integration Transformer (HMIT) in simulating physical systems, its initial implementation faced notable limitations in computational speed and memory consumption. To address these challenges, we incorporated two key optimizations. First, **KV Cache** eliminates redundant computations by caching key and value matrices, reducing the attention complexity from $O(3Nsd^2 + 3Nsd)$ to $O(Nsd^2 + 3Nsd)$. Second, **Dynamic Weighted Value Selection** dynamically selects the first $m$ rows of the weighted value matrix $w \in \mathbb{R}^{s \times d}$, where $m$ corresponds to the current message-passing step, further enhancing computational efficiency. As shown in the above table, these optimizations significantly reduce training and inference times while decreasing memory requirements.

---

### Official Review · Reviewer_wc4g · 2024-10-31

**Soundness:** 3
**Presentation:** 2
**Contribution:** 3
**Rating:** 5
**Confidence:** 4

**Summary:**

The paper introduces the Message Passing Transformer for physical simulations. MPT employs an Encoder-Processor-Decoder framework and integrates two innovative components: Hadamard-Product Attention and Graph Fourier Loss. Hadamard-Product Attention assigns attention weights at the feature level rather than by sequence position. Graph Fourier Loss is introduced as a spectral loss function that balances high- and low-energy graph components. By precomputing the graph’s Laplacian eigenvectors, GFL maintains computational efficiency, as this precomputation does not impact inference time.

**Strengths:**

- The proposed Hadamard-Product Attention offers a fine-grained approach to attention by assigning weights to each feature dimension. The experiments show that it brings an improvement over traditional Dot-Product Attention.
- The application of Graph Fourier Loss to balance spectral components is novel, leveraging graph signal processing to enhance model accuracy over extended rollouts in physical simulations.

**Weaknesses:**

- The computational requirements of the proposed method are significantly greater than all the baselines. Moreover, the computational cost for precomputing Laplacian eigenvectors is not discussed.
- From my understanding, the precomputation of Laplacian eigenvectors is not feasible for dynamic graphs that undergo frequent topological changes, such as some of the datasets regarding dynamic flags proposed in the MGN paper.
- The ablation study in Table 2 is only conducted on the CylinderFlow dataset.
- In Figure 3(a), is there something wrong with the label on the graph? It's quite confusing that the figure aims to show the effectiveness of Hadamard-Product Attention while it compares the results of different segment rate.
- In Figure 8, it would be better if the authors could also present the ground truth results for better comparsion.
- Typo: Line 481, "wwe"

**Questions:**

- What does "weight of the edge" mean in Line 131?
- How is the weighted value matrix $w$ reshaped back to a dimension of d in detail? Is the process implemented with a learned MLP?
- During each step of message passing, are the networks $f_1, f_2, f_3, f_4, f_5$ shared or independent, respectively?

---

> ### Author Response · Authors · 2024-11-21
> **Individual response**
>
> Thank you for your insightful comments and constructive feedback. Below, we provide detailed responses to each point raised:
>
> **1. The computational cost for precomputing Laplacian eigenvectors**
> **Response**:
>
> **[Experiment: Precomputation Cost]**:
> | Dataset   | Eigen Decomposition Time per Sample| Total Processing Time  |
> |-----------|------------------------------------|------------------------|
> | Cylinder  | 0.026 s                            | 27.21 s                |
> | Airfoil   | 0.204 s                            | 218.34 s               |
> | Plate     | 0.016 s                            | 19.19 s                |
> | Flag      | 0.020 s                            | 21.82 s                |
>
> While we provide a precomputation cost table for clarity, this data processing only occurs once and can be considered part of the dataset generation process.
>
> The primary computational cost is in the eigen decomposition of the graph Laplacian, with a total precomputation time of about 3 minutes—negligible compared to training time. For the four datasets with static graph topologies, eigen decomposition is only needed at $t=0$ rather than at each time step.
>
> For datasets with dynamic topologies, eigen decomposition is required at each time step but only needs to be done once, in batch, before training, and cached for reuse. This approach reduces on-the-fly calculation complexity, improving training efficiency. Importantly, inference does not require precomputed eigenvalues or eigenvectors, so inference speed remains unaffected.
>
> **2. The precomputation of Laplacian eigenvectors is not feasible for dynamic graphs with frequent topology changes**
> **Response**: Thank you for pointing this out. We clarify that our GFL **can** be applied to cases with changing graph topology, as long as GFT is performed during preprocessing for each time step. Relevant sections of the paper have been revised accordingly.
>
> **3. The ablation study in Table 2 is limited to the CylinderFlow dataset.**
> **Response**:
>
> **[Experiment: ablation in FlagSimple]**
> | Measurements    | Without HPA and GFL | HPA only | GFL only | HPA + GFL (Ours) |
> |-----------------|---------------------|----------|----------|------------------|
> | RMSE-1 [1e-2]   | 6.47E-2             | 1.51E-2  | 2.29E-2  | **1.12E-2**      |
> | RMSE-50 [1e-2]  | 2.29                | 1.97     | 2.03     | **1.87**         |
> | RMSE-all [1e-2] | 2.45                | 2.16     | 2.21     | **2.05**         |
>
> In the ablation study focusing on the FlagSimple dataset, the integration of both HPA and GFL demonstrated the highest improvement in reducing error rates across all RMSE measures.
>
> **4. In Figure 3(a), the graph label is confusing, especially for demonstrating Hadamard-Product Attention effectiveness.**
> **Response**: Thank you for pointing this out. We apologize for the annotation mistake; the gray line represents Dot-Product Attention, while the red line represents our Hadamard-Product Attention. We have corrected this in the paper accordingly.
>
> **5. What is meant by "weight of the edge" in Line 131?**
> **Response**: We have modified the paper’s wording to make it clearer: The adjacency matrix of $G$ is denoted by $A$, where $A_{ij} = 1$ if there is an edge between vertices $i$ and $j$, and $A_{ij} = 0$ otherwise.
>
> **6. How is the weighted value matrix 𝑤 reshaped back to 𝑑 dimensions, and is an MLP used for this?**
> **Response**: The weighted value matrix $w \in \mathbb{R}^{d \times s}$ is reshaped to $d$ dimensions via a learned linear layer $W_{\text{linear}} \in \mathbb{R}^{s \times 1}$, applying a transformation $w W_{\text{linear}}$. This is a simple linear layer without non-linear activation, not an MLP.
>
> **7. During each step of message passing, are the networks $f_1, f_2, f_3, f_4, f_5$ shared or independent?**
> **Response**: The five networks are independent of each other.
> * $f_1$ and $f_2$ take node features (e.g., velocity) and edge features (e.g., length) as inputs, respectively, and output 128-dimensional vectors.
> * $f_3$ is an independent MLP for each MP layer. For example, if $\text{MP} = 15$, then $f_3$ generates 15 MLPs, each with an input size of 128, an output size of 128.
> * $f_4$ uses the same weights across different MP layers and implements Multihead Hadamard-Product Attention.
> * $f_5$ decodes the 128-dimensional node latent features back to the real space.

---

> ### Author Response · Authors · 2024-11-25
> **Reminder: Public Discussion Period for Submission 2831**
>
> Dear reviewer, this is a friendly reminder that the public discussion period for Submission 2831 is ending soon. If you have any questions or concerns about our rebuttal, we’d be happy to address them. Thank you for your time and feedback!

---

> ### Comment · Reviewer_wc4g · 2024-11-26
>
> Thank you for your answers.
>
> Regarding W1. Can you analyze the computational complexity when using Hadamard-Product Attention compared to regular message-passing and Dot-Product Attention?
>
> Regarding Table 1. Why is there a huge discrepancy in the results of MGN and BSMS compared to the results in their original paper? One may even find some results of baselines in their original paper are better than the results of the proposed MPT (eg. RMSE-all of MGN on Airfoil and BSMS on Plate).
>
> I also recommend that the authors upload the revised paper to address some concerns in the rebuttal.

---

> > ### Author Response · Authors · 2024-11-26
> > **Response to Comment by Reviewer**
> >
> > **1. Regarding W1. Can you analyze the computational complexity when using Hadamard-Product Attention compared to regular message-passing and Dot-Product Attention?**
> > **Response:** The computational complexity of our HPA (now renamed as Historical Message-Passing Attention, HMPA) is essentially similar to that of Dot-Product Attention. For regular message-passing (e.g., MGN), the computational complexity is approximately $O(d^2 \cdot (N+E))$, where $d$ is the hidden layer dimension, and $N$ and $E$ represent the numbers of nodes and edges, respectively. It is worth noting that in typical  graph structures, the number of edges $E$ is usually much larger than the number of nodes $N$. However, HPA performs computations only on the nodes rather than on the edges, which significantly reduces the computational overhead. Even in cases where the number of edges is large, HPA remains computationally efficient and within acceptable runtime limits.
> >
> > **Complexity Analysis**
> >     We calculate the FLOPs for the HPA mechanism, with $N$ as the number of nodes, $s$ as the sequence length, and $d$ as the feature dimension. Constructing the key and value matrices $K$ and $V$ requires linear transformations across each sequence step, amounting to $2Nsd^2$ FLOPs. The calculation of attention weights $a$, including the element-wise product and softmax operation, requires $2Nsd$ FLOPs. The element-wise multiplication of $a$ and $V$, followed by a linear transformation to reshape the result back to dimension $d$, adds $Nsd + Nsd^2$ FLOPs. Summing these components, the total FLOPs for HPA is $3Nsd^2 + 3Nsd$. This complexity scales **linearly** with the number of nodes $N$, with a fixed sequence length $s$. Compared to a global self-attention mechanism, our decoder-only architecture avoids the $O(N^2)$ complexity, making it especially efficient and advantageous for large-scale simulations.
> >
> > **2. Regarding Table 1. Why is there a huge discrepancy in the results of MGN and BSMS compared to the results in their original paper? One may even find some results of baselines in their original paper are better than the results of the proposed MPT (eg. RMSE-all of MGN on Airfoil and BSMS on Plate).**
> > **Response**: We utilized the [`mgn_pytorch`](https://github.com/echowve/meshGraphNets_pytorch) implementation to run MGN baseline, yielding the results shown in the table. We note that other papers using MGN's datasets[1][2][7] also show numbers that are quite substantially different. The BSMS result for the plate is smaller by an order of magnitude compared to the original MGN result. We believe this discrepancy is caused by differences in certain settings.
> >
> > | **Measurements** | **Dataset** | **Original MGN Result** | **Our's MGN Result** | **BSMS's MGN [7]** | **Mesh Transformer's MGN [1]** | **CARE's MGN [2]** | **BSMS[7]** |
> > |-------------------|-------------|--------------------------|-----------------------|--------------------|---------------------------------|--------------------|----------|
> > | **RMSE-1**       | Cylinder    | 0.00234                 | 0.00524              | 0.00226            | 0.0058                          | 0.1434            | 0.00204   |
> > |                  | Airfoil     | 0.314                   | 3.14                 | 0.435              | -                               | 0.0385            | 0.288    |
> > |                  | Plate       | 0.00025                 | 0.000269             | 0.000198           | -                               | -                 | 0.000287  |
> > | **RMSE-50**      | Cylinder    | 0.0063                  | 0.014                | 0.0439             | 0.0405                          | 0.4328            | 0.0242  |
> > |                  | Airfoil     | 0.582                   | 5.36                 | 16.6               | -                               | 0.0751            | 0.605    |
> > |                  | Plate       | 0.0018                  | 0.00173              | 0.000288           | -                               | -                 | 0.0019   |
> > | **RMSE-all**     | Cylinder    | 0.04088                 | 0.0432               | 0.107              | -                               | -                 | 0.0837    |
> > |                  | Airfoil     | 11.529                  | 20.8                 | 69.6               | -                               | -                 | 42.1    |
> > |                  | Plate       | 0.0151                  | 0.0161               | 0.00151            | -                               | -                 | 0.0016    |
> >
> > Once we have confirmed that all reviewers' requirements have been met, we will immediately upload the revised paper.

---

> > ### Author Response · Authors · 2024-11-27
> > **The revised paper has been uploaded**
> >
> > Thank you for your thoughtful feedback. We truly appreciate your time and consideration. We would like to inform you that the revised version of the paper has now been uploaded. For a detailed overview of the changes, you can refer to the latest overall official comment.

---

### Official Review · Reviewer_L6SR · 2024-11-05

**Soundness:** 3
**Presentation:** 3
**Contribution:** 2
**Rating:** 6
**Confidence:** 5

**Summary:**

The authors introduce the message passing transformer, a graph neural network transformer hybrid introducing two novel components, the Hadamard product attention, and a novel graph Fourier loss, to improve the algorithm's training speed the Laplacian eigenvectors are precomputed. Rollouts of the spatiotemporal dynamics are tested on the datasets of the Meshgraphnet paper, with long-term rollouts being found to be more stable across Eulerian, as well as Lagrangian dynamics.

**Strengths:**

Where the paper shines is the theoretical background of its approach, the entire model architecture, as well as formulation intricacies are described in detail, and allow for an in-depth look at the architecture, the notable differences to preceding work like the Hadamard-Product attention, and the graph Fourier loss. As far as the reviewer can tell, the architecture should be fully reproducible from this exhibition.

**Weaknesses:**

Where this paper falls short in its current form is the evaluation, and its embedding into present literature.

Maybe slightly too focussed on PINN literature, it misses two landmark works of the past year:

* The Universal Physics Transformers of Alkin et al., which also have a GNN-core and hence fall squarely into the category of a GNN-Transformer hybrid like the presented architecture
* Poseidon: Efficient Foundation Models for PDEs by Herde et al., which is also built to spatio-temporally evolve physical systems These two works, as well as the datasets they were introduced with are presently not represented in the paper.

It would improve the related work greatly to contrast these approaches to highlight the architectural differences, and if on purpose to make sure that it is clear to the reader why the proposed architecture is not compared to the previous two. Both models are freely available, and could be fine-tuned to the datasets used in this paper. Leading into the evaluation, the reviewer would furthermore question whether the models evaluated against are representative of the current state-of-the-art.

There have been a number of new datasets since MGN, not only the two previously mentioned works, but furthermore other datasets for PDEs/Physical systems which are more challenging than MGN's datasets. I would urge the authors to consider extending their evaluation to these more recent datasets. Concretely, the following datasets could provide a good starting point:
* _PDEBENCH: An Extensive Benchmark for Scientific Machine Learning_ by Takomoto et al.
* _LagrangeBench: A Lagrangian Fluid Mechanics Benchmarking Suite_ by Toshev et al.
* _PINNacle: A Comprehensive Benchmark of Physics-Informed Neural Networks for Solving PDEs_ by Hao et al. includes benchmarks simulated with spectral methods, which should lend itself to the style of the message-passing Transformer.
* _Toward multi-spatiotemporal-scale generalized PDE modeling_ by Gupta et al.

More directly with the two previously mentioned models, I would urge the reviewers to consider extending their evaluations to more related models to allow for a better comparison against contemporary work. Specifically for the problem of long rollouts in the context of machine learning models for particle dynamics, _Neural SPH: Improved Neural Modeling of Lagrangian Fluid Dynamics_ was introduced at ICML'24 by Toshev et al., alongside its datasets, and would lend itself to comparison. It is unclear to the reviewer why the proposed architecture was not compared to this state-of-the-art architecture for longterm rollouts of particle dynamics. Either comparing to this architecture, or

**Questions:**

Seeing that the entire model is trained, as well as inferred on a single RTX 4090, do you see a way to compare the performance against the other evaluated models across training compute, as well as the cost of inference? As is, it feels like this computational efficiency is not captured properly by the evaluation design. Specific comparisons I think would help to emphasize this point would be

* Rollout performance on a fixed training budget on aforementioned RTX 4090. If you were to e.g. take only the training budget of the introduced model, and retrain the compared to models with only this budget, how would rollout performance compare?
* At inference time, one could for example evaluate the time to first sample as a comparative metric on the chosen hardware, my intuition woulds be that the optimizations done by you with the precomputation & caching should aid your proposed approach considerably.

---

> ### Author Response · Authors · 2024-11-21
> **Individual response Part 1.**
>
> Thank you for your insightful comments and constructive feedback. Below, we provide detailed responses to each point raised:
>
> **1. Additional landmark works**
> **Response**: Thank you for your suggestion. We have included **The Universal Physics Transformers** by Alkin et al. and **Poseidon: Efficient Foundation Models for PDEs** by Herde et al. as additional citations, to acknowledge recent foundational work in this area. As our model is a GNN-based architecture, we have avoided direct comparisons with non-GNN-based methods. However, we have added three additional GNN-based baselines in response to the question below.
>
> **2. The paper's evaluation and literature integration are lacking.**
> **Response**:
> **[Experiment: Additional Dataset Test]**:
>
> | Measurements    |Ours (MPT)    | MGN       | BSMS        | TIE       |
> |-----------------|--------------|-----------|-------------|-----------|
> | RMSE-1          | **1.08 E-01**| 2.25 E-01 | 1.87 E-01   | 1.62 E-01 |
> | RMSE-50         | **2.57 E-01**| 6.03 E-01 | 5.34 E-01   | 5.71 E-01 |
> | RMSE-all (100)  | **4.63 E-01**| 9.25 E-01 | 8.36 E-01   | 8.49 E-01 |
>
> We additionally conducted an experiment on the Dam Problem from the 2023 CFDBench [2] benchmark, which models the rapid release of water from a column collapse and represents complex free-surface flows with varying velocities. Our method outperforms previous methods (MGN, BSMS, and TIE) across all three RMSE metrics—RMSE-1, RMSE-50, and RMSE-all—by 33%-57%.
>
>  **[Experiment: Additional Baslines Comparison]**:
>
> | Measurements    | Dataset   | Ours (MPT)         |Mesh Transformer[2]| HCMT[3]          |Graph MLP-MIXER[4]|
> |-----------------|-----------|--------------------|-------------------|------------------|------------------|
> | RMSE-1 [1E-2]   | Cylinder  | **2.03 E-01**      | 3.05 E-01         | -                | 4.12 E-01        |
> |                 | Airfoil   | **2.61 E+02**      | 2.96 E+02         | -                | 3.05 E+02        |
> |                 | Plate     | **1.00 E-02**      | 2.38 E-02         | 1.15 E-02        | 3.28 E-02        |
> |                 | Flag      | **1.12 E-02**      | 4.73 E-02         | -                | 6.89 E-02        |
> | RMSE-50 [1E-2]  | Cylinder  | **6.32 E-01**      | 1.07              | -                | 4.61             |
> |                 | Airfoil   | **4.08 E+02**      | 5.21 E+02         | -                | 6.32 E+02        |
> |                 | Plate     | **9.25 E-02**      | 1.30 E-01         | 9.68 E-01        | 6.19 E-01        |
> |                 | Flag      | **1.87**           | 2.04              | -                | 3.90             |
> | RMSE-all [1E-2] | Cylinder  | **3.78**           | 4.26              | -                | 2.05 E+01        |
> |                 | Airfoil   | **1.64 E+03**      | 2.00 E+03         | -                | 3.97 E+03        |
> |                 | Plate     | **1.09**           | 1.28              | 1.15             | 8.22             |
> |                 | Flag      | **2.05**           | 2.32              | -                | 7.25             |
>
> Currently, we use state-of-the-art message-passing GNNs(MGN, BSMS) and Transformer-based method(TIE) as our baselines. However, to enable a more comprehensive comparison, we have added Mesh-Transformer[1] and Graph MLP-MIXER[3] as additional baselines for testing. We additionally test [2] only on DeformingPlate dataset because its CMT and HMT modules were ultimately designed specifically for collision cases.

---

> > ### Author Response · Authors · 2024-11-21
> > **Individual response Part 2.**
> >
> > **3. How would rollout performance compare if baseline models were retrained with the same budget on an RTX 4090?**
> > **Response**:
> > * Our model and baselines in the paper are compared with similar parameter scales, each having a hidden layer dimension of 128 and a roughly equivalent parameter count.
> > * Additionally, we trained and tested MGN, BSMS, and TIE with a hidden layer dimension of 256 on the DeformingPlate dataset, resulting in a training budget slightly higher than our method. The results show only minor performance improvements and still fall short of achieving the accuracy of our model (with 128 dimensions).
> >     **[Experiment: Runtime Consistency Performance]**
> >
> >     | Measurements    | Ours (MPT)    | MGN (128)  | MGN (256)  | BSMS (128)  | BSMS (256)  | TIE (128)  | TIE (256)  |
> >     |-----------------|---------------|------------|------------|-------------|-------------|------------|------------|
> >     | RMSE-1          | **1.00 E-02** | 2.69 E-02  | 2.14 E-02  | 2.83 E-02   | 2.65 E-02   | 3.56 E-02  | 3.21 E-02  |
> >     | RMSE-50         | **9.25 E-02** | 1.73 E-01  | 1.62 E-01  | 2.81 E-01   | 2.43 E-01   | 3.61 E-01  | 3.37 E-01  |
> >     | RMSE-all (100)  | **1.09**      | 1.61       | 1.52       | 4.52        | 4.34        | 9.62       | 9.05       |
> >
> >
> > **4. For inference, it may be useful to evaluate the time to first sample on the selected hardware. Precomputation and caching optimizations should help here.**
> > **Response**: As mentioned in our paper, our Graph Fourier Loss (GFL) is only used during training, and the precomputation of the graph Laplacian primarily accelerates the training process, with no impact on inference speed. We have added the precomputation time for the graph Laplacian. Note that the training time per step for our model is 0.008 seconds. Without precomputation, the training time would increase significantly.
> > **[Experiment: Precomputation Cost]**:
> >
> > | Dataset   | Eigen Decomposition Time per Sample| Total Processing Time  |
> > |-----------|------------------------------------|------------------------|
> > | Cylinder  | 0.026 s                            | 27.21 s                |
> > | Airfoil   | 0.204 s                            | 218.34 s               |
> > | Plate     | 0.016 s                            | 19.19 s                |
> > | Flag      | 0.020 s                            | 21.82 s                |
> >
> >
> > [1] CFDBench: A Comprehensive Benchmark for Machine Learning Methods in Fluid Dynamics
> > [2] Eagle: Large-scale learning of turbulent fluid dynamics with mesh transformers
> > [3] Learning flexible body collision dynamics with hierarchical contact mesh transformer
> > [4] A GENERALIZATION OF VIT/MLP-MIXER TO GRAPHS

---

> > > ### Comment · Reviewer_L6SR · 2024-11-25
> > > **Response to Rebuttal**
> > >
> > > I thank the authors for their very extensive rebuttal, and addition of further evaluations. Having read all responses to the reviewers, as well as the general response to reviewers, I have raised my review's score accordingly.

---

### Official Review · Reviewer_G8Yr · 2024-11-06

**Soundness:** 3
**Presentation:** 3
**Contribution:** 3
**Rating:** 6
**Confidence:** 4

**Summary:**

This work proposes a new hybrid Graph Neural Network based autoregressive Transformer architecture. The authors use an Encoder-Processor-Decoder (EPD) architecture to model physical simulations.
The major contributions of the work include the scaled Hadamard Product Attention (along feature dimensions) and a novel loss function which utilises Graph Fourier Transform.
The authors have shown the effectiveness of the proposed model on both eulerian and lagrangian dynamical systems by achieving superior performance over previous GNN and Transformer based methods.

**Strengths:**

S1) Novel Graph Fourier Loss which helps learn complex physical phenomena effectively across the energy spectrum of the system. This helps avoid using the Graph Fourier Transform in both training and inference

S2) Modified Attention mechanism focusing on obtaining importance of features as scores by using softmax along the dimension and not along the sequence. This is one of the major contributors to the results as the graph structure helps with message passing/interaction and hence softmax can be applied along the dimension instead of sequence.

S3) Detailed experiments and theoretical/mathematical justification alongwith ablations of all the design choices considered. The combination of HPA and GFL seems to achieve superior performance by learning in both spatial and spectral domains.

**Weaknesses:**

W1) Comparison between the current method and previous methods in terms of the wall clock time and memory footprint have not been included (The authors have mentioned this as a limitation and also in section F of the appendix). A thorough quantitative analysis of time and memory for each component would be useful for the current and future research as well.

W2) Although the GFL seems very effective, how one arrives at the formulation is not very clear. How does one arrive at the expression of $\alpha$ and and the concept of segment rate as well  in section 4.3 is not straightforward. This makes it unclear of how to build further on this work.

**Presentation/Typos/Corrections**

Section 1 line 47-48
*we replace the commonly used Multi-Layer Perceptron (MLP) with Hadamard-Product Attention.*
From what I understand in Figure 1, we are still having MLP (FFN) layers right. Should this not be *we replace the commonly used Scaled Dot Product Attention with Hadamard-Product Attention.* instead?

In Figure 1, I believe it should be *xM* times instead of *xMP* times as MP is not a quantity and refers to Message Passing

Section 2 line 63 *methods have achieved* or *methods achieve* instead of *methods have achieve*

Section 3 line 116-117 For the clarity of notation, can we have a superscript $t$ to denote time along with the currently defined iteration $k$ as a subscript. I feel this will greatly enhance readability (Also make the changes elsewhere)

Section 4.2.1 line 257-259 Is the expression for the gradient computation correct? I believe either a parenthesis is missing or the $\sigma$ should be moved after the first term.

**Questions:**

Q1) What does MHHA refer to in the figure 1, I believe it is the Scaled Hadamard Product Attention (HPA)?

Q2) Can this model be used effectively with the pretrain and then finetune paradigm (especially on completely different equations/regimes) ? How generalizable is the model across equations, regimes, meshes etc?

Q3) How would this GNN-Transformer modelling of the system behave if we wanted to incorporate physics-informed loss components (Especially as the loss is GFL and not standard MSE)?

---

> ### Author Response · Authors · 2024-11-21
> **Individual response Part 1.**
>
> Thank you for your insightful comments and constructive feedback. Below, we provide detailed responses to each point raised:
>
> **1. Wall clock time and memory comparisons are missing. A detailed time and memory analysis per component would aid future research.**
> **Response**: We have noted this as a limitation in the paper and also discussed it in section F of the appendix. Additionally, we have included a graph Laplacian precomputation cost table and replace "**Gradient Computation in HPA**" section with "**Complexity Analysis**" in the paper.
> **[Experiment: Precomputation Cost]**:
>
> | Dataset   | Eigen Decomposition Time per Sample| Total Processing Time  |
> |-----------|------------------------------------|------------------------|
> | Cylinder  | 0.026 s                            | 27.21 s                |
> | Airfoil   | 0.204 s                            | 218.34 s               |
> | Plate     | 0.016 s                            | 19.19 s                |
> | Flag      | 0.020 s                            | 21.82 s                |
>
> While we provide a precomputation cost table for clarity, this data processing only occurs once and can be considered part of the dataset generation process.
>
> The primary computational cost is in the eigen decomposition of the graph Laplacian, with a total precomputation time of about 3 minutes—negligible compared to training time. For the four datasets with static graph topologies, eigen decomposition is only needed at $t=0$ rather than at each time step.
>
> For datasets with dynamic topologies, eigen decomposition is required at each time step but only needs to be done once, in batch, before training, and cached for reuse. This approach reduces on-the-fly calculation complexity, improving training efficiency. Importantly, inference does not require precomputed eigenvalues or eigenvectors, so inference speed remains unaffected.
>
> **Complexity Analysis**
>     We calculate the FLOPs for the HPA mechanism, with $N$ as the number of nodes, $s$ as the sequence length, and $d$ as the feature dimension. Constructing the key and value matrices $K$ and $V$ requires linear transformations across each sequence step, amounting to $2Nsd^2$ FLOPs. The calculation of attention weights $a$, including the element-wise product and softmax operation, requires $2Nsd$ FLOPs. The element-wise multiplication of $a$ and $V$, followed by a linear transformation to reshape the result back to dimension $d$, adds $Nsd + Nsd^2$ FLOPs. Summing these components, the total FLOPs for HPA is $3Nsd^2 + 3Nsd$. This complexity scales **linearly** with the number of nodes $N$, with a fixed sequence length $s$. Compared to a global self-attention mechanism, our decoder-only architecture avoids the $O(N^2)$ complexity, making it especially efficient and advantageous for large-scale simulations.
>
> **2.  The derivation of GFL, including the expression of $\alpha$ and the segment rate $s_r$, is unclear, making it difficult to extend.**
> **Response**:
> * The concept of the segment rate $s_r$ was introduced to divide the frequency domain energy into high-energy and low-energy components, ensuring that the model focuses on the most relevant frequency components during training.
> * The adjustment factor $\alpha$, derived from the ratio of mean energy in high- and low-energy partitions, adaptively balances their contributions to the loss.
> $$
> \alpha = \sqrt{\frac{\text{mean}(E_{\text{high}})}{\text{mean}(E_{\text{low}}) + \epsilon}} \cdot \lambda
> $$
> * Without the adjustment factor $\lambda$, $\alpha$ would excessively boost low-energy components to match high-energy levels, amplifying noise and reducing performance. Introducing $\lambda$ allows adaptively controlled scaling in low-energy regions.
>
> **3. Section 1, line 47-48: Should the phrase about replacing MLP with Hadamard-Product Attention instead reference replacing Scaled Dot Product Attention?**
> **Response**: Thank you for your feedback.
> * In lines 47-48, "MLP" refers to using MLPs in each Message Passing step to model node-edge interactions, similar to MGN and GNS, where concatenated node and edge features are input and the next step’s node features are output.
> * In our architecture, however, the FFN applies non-linearity to HPA-modulated node information, with nodes as both input and output.

---

> > ### Author Response · Authors · 2024-11-21
> > **Individual response Part 2.**
> >
> > **4. Section 3, line 116-117: Can notation be clarified with a superscript $t$ for time along with subscript $k$ for iteration?**
> > **Response**: Thank you for your suggestion. We have revised the wording in both the Problem Formulation section and the subsequent parts of the paper: We consider graph $G^t = (V^t, E^t)$ to represent a physical system with $t$ taking discrete values $t = 0, 1, \ldots$, where $V^t$ denotes the set of nodes with node attributes $v^t_i$ for each $v^t_i \in V^t$, and $E^t$ denotes the set of edges with edge attributes $e^t_{ij}$ for each $e^t_{ij} \in E^t$. This implies a constant structure where the connectivity pattern among nodes is preserved across all time steps. We also define a total of $M$ Message Passing iterations, with $k = 0, 1, \ldots, M$. During the $k$-th Message Passing iteration, the attributes of nodes and edges are denoted by $v^t_{k,i}$ and $e^t_{k,ij}$.
> >
> > **5. Section 4.2.1, line 257-259: Is the gradient expression correct? A parenthesis might be missing or $\sigma$ misplaced.**
> > **Response**: Thank you for your suggestion. The original notation might have caused some ambiguity, so we have added parentheses to clarify that the summation is over the entire product term:
> > $$
> > \frac{\partial \mathcal{L}}{\partial v_q} = \sum_{p=1}^{s} \left( \frac{\partial \mathcal{L}}{\partial a_{p,q}} \cdot \frac{\partial a_{p,q}}{\partial v_q} \right).
> > $$
> > But we finally decided to replace this section with "**Complexity Analysis**", because the gradient equation may not be important.
> >
> > **6. What does MHHA refer to in the figure 1?**
> > **Response**: MHHA refer to Multi-head Hadamard-Product Attention, we have decided to rename Hadamard-Product Attention (HPA) to Historical Message-Passing Attention (HMPA). Accordingly, all references to HPA in the figures of the paper have been replaced with Historical Message-Passing Attention (HMPA).
> >
> > **7.  Is the model suitable for a pretrain-finetune approach, especially for different equations or regimes? How well does it generalize?**
> > **Response**: Thank you for the insightful question.
> > * Due to hardware constraints (a single 4090 GPU), our model is a smaller version tailored to specific equations without fine-tuning, consistent with GraphNeuralSimulator, MeshGraphNet, Bi-Stride MultiScale-GNN, TIE, Mesh Transformer[1], and CARE[2]. Most current methods target specific problems, as large models supporting pre-training and fine-tuning demand extensive hardware resources, placing them in a different category.
> > * Our Message Passing Transformer model demonstrates strong generalization, as the dataset we use is the same as that in MGN, encompassing different system parameters(2D & 3D, density & pressure & momentum & von-Mises stress ...), mesh shapes(triangle & tetrahedra, regular & irregular), and mesh sizes(airfoil dateset edge lengths range between $2·10^{-4}$m to 3.5m).
> >
> >
> > **8. How would the model handle physics-informed loss components, given GFL not MSE loss?**
> > **Response**: Thank you for bringing up an important question.
> > * One straightforward method is to perform an inverse Graph Fourier Transform (GFT) on the Graph Fourier Loss after calculation, converting the loss back to the spatial domain. This transformed GFL can then be combined with a physical loss in the spatial domain, forming the final total loss.
> > * Another method is to define the physical constraints in the spatial domain and then transform them into the frequency domain via GFT, allowing for direct comparison with the model's frequency-domain output. In the frequency domain, the same or separate adjustment factors can be used to modulate the physical constraints.
> >
> > [1] Eagle: Large-scale learning of turbulent fluid dynamics with mesh transformers
> > [2] CARE: Modeling Interacting Dynamics Under Temporal Environmental Variation

---

> ### Author Response · Authors · 2024-11-25
> **Reminder: Public Discussion Period for Submission 2831**
>
> Dear reviewer, this is a friendly reminder that the public discussion period for Submission 2831 is ending soon. If you have any questions or concerns about our rebuttal, we’d be happy to address them. Thank you for your time and feedback!

---

> > ### Comment · Reviewer_G8Yr · 2024-11-26
> >
> > I have read through all the reviewers comments and author rebuttals and responses. I am satisfied with the responses. I would like to thank the authors for extensive additional experiments and clarifications provided which have improved this work greatly. I have no further questions, clarifications or requirements. I will retain the scores.

---

> ### Author Response · Authors · 2024-11-28
> **Consideration of Further Suggestions for Improving the Score**
>
> Thank you for your thoughtful review and valuable feedback. We appreciate the time you have taken to evaluate our work. We would like to ask if there is anything else we can do to improve the rating of our submission. Your insights would be greatly appreciated as we continue to refine our approach.
>
> In our discussions with Reviewer ZnqH, we further optimized the computational efficiency of our method, significantly reducing the increase in time and memory usage compared to MGN. This means that the wall clock time and memory footprint of our model are **no longer limiting factors**. As shown in the table below, the optimized version of HMIT achieves substantial improvements in both training and inference time, while also reducing memory usage:
>
> | Measurements                             | MGN | HMIT (ours) | optimized HMIT (ours) |
> |------------------------------------------|-----------|-------------|-----------------------|
> | $t_\text{train} / \text{step} \, [\text{ms}]$ | 4.59      | 8.12        | 5.24                  |
> | $t_\text{infer} / \text{step} \, [\text{ms}]$ | 1.91      | 3.37        | 1.95                  |
> | Train RAM [GB]                           | 1.50      | 4.45        | 1.92                  |
> | Infer RAM [GB]                           | 0.61      | 0.79        | 0.64                  |
>
> Despite the promising advancements offered by the Historical Message-Passing Integration Transformer (HMIT) in simulating physical systems, its initial implementation faced notable limitations in computational speed and memory consumption. To address these challenges, we incorporated two key optimizations. First, **KV Cache** eliminates redundant computations by caching key and value matrices, reducing the attention complexity from $O(3Nsd^2 + 3Nsd)$ to $O(Nsd^2 + 3Nsd)$. Second, **Dynamic Weighted Value Selection** dynamically selects the first $m$ rows of the weighted value matrix $w \in \mathbb{R}^{s \times d}$, where $m$ corresponds to the current message-passing step, further enhancing computational efficiency. As shown in the above table, these optimizations significantly reduce training and inference times while decreasing memory requirements.

---

### Author Response · Authors · 2024-11-21
**General response Part 1.**

## General response:
We thank all reviewers and the AC for their time and effort in reviewing our work and for providing insightful comments to strengthen it. We have addressed typographical and grammatical errors throughout the paper, as noted by [Reviewer G8Yr, Reviewer wc4g]. We have also updated figure labels and representations to improve clarity and accuracy, as suggested by [Reviewer G8Yr, Reviewer wc4g, Reviewer YBDW]. Additionally, we have added visualizations of ground truth results to enhance experimental clarit [Reviewer wc4g]. **If further experimental evidence, including additional open-source baselines or datasets, is required, we will provide these as soon as possible.**

1. **Contributions**
    1. **[Motivation]**
        * To mitigate over-squashing, capture fine-grained details, and scale linearly with node count, we purpose HPA.
        * To modulate loss at specific frequencies and handle varying energy levels, we introduce GFL.
    3. **[Method]**
        * Our Hadamard-Product Attention (HPA) integrates multi-step historical message-passing information with feature-wise softmax and a decoder-only architecture.
        * Our Graph Fourier Loss (GFL) uses a frequency-domain energy adjustment schedule, similar to a piecewise function.
    4. **[Experiments]** Extensive experiments and ablations show that HPA and GFL boost performance across datasets, outperforming traditional attention mechanisms [Reviewer wc4g, Reviewer YBDW].
    5. **[Presentation]** Our manuscript includes detailed architectural and theoretical explanations, ensuring reproducibility and clarifying innovations[Reviewer L6SR].


2. **New Results**
    1. **[Method: Complexity Analysis]**. We have modified the paper and replaced the section '**Gradient Computation in HPA**' with '**Complexity Analysis**' [Reviewer G8Yr, Reviewer YBDW].
    2. **[Experiment: Precomputation Cost]**. We presented the precomputation time for the graph Laplacian matrix [Reviewer G8Yr, Reviewer L6SR, Reviewer wc4g, Reviewer YBDW, Reviewer ZnqH].
    3. **[Experiment: Ablation in FlagSimple]**. We tested the individual and combined performance of model components on the FlagSimple dataset [Reviewer wc4g].
    4. **[Experiment: Additional Baselines Comparison]**. We additionally ran the Mesh Transformer[1], HCMT[3], Graph MLP-MIXER[4] to demonstrate the superiority of our method [Reviewer L6SR, Reviewer YBDW, Reviewer ZnqH].
    5. **[Experiment: Additional Dataset Test]**. We selected the Dam Problem from CFDBench [2] as an additional dataset for performance comparison [Reviewer L6SR].
    6. **[Experiment: Runtime Consistency Performance]**. We tested runtime performance for models with varying hidden dimensions (128 and 256) on the DeformingPlate dataset [Reviewer L6SR].
    7. **[Experiment: GFL Robustness Performance]**. We evaluated energy changes after randomly removing different proportions of edges on the Airfoil dataset to demonstrate the robustness of GFL to graph topology variations [Reviewer ZnqH].

3. **Complexity Analysis[Reviewer G8Yr, Reviewer YBDW]**
    We calculate the FLOPs for the HPA mechanism, with $N$ as the number of nodes, $s$ as the sequence length, and $d$ as the feature dimension. Constructing the key and value matrices $K$ and $V$ requires linear transformations across each sequence step, amounting to $2Nsd^2$ FLOPs. The calculation of attention weights $a$, including the element-wise product and softmax operation, requires $2Nsd$ FLOPs. The element-wise multiplication of $a$ and $V$, followed by a linear transformation to reshape the result back to dimension $d$, adds $Nsd + Nsd^2$ FLOPs. Summing these components, the total FLOPs for HPA is $3Nsd^2 + 3Nsd$. This complexity scales **linearly** with the number of nodes $N$, with a fixed sequence length $s$. Compared to a global self-attention mechanism, our decoder-only architecture avoids the $O(N^2)$ complexity, making it especially efficient and advantageous for large-scale simulations.


4. **[Experiment: Precomputation Cost]**[Reviewer G8Yr, Reviewer L6SR, Reviewer wc4g，Reviewer YBDW, Reviewer ZnqH]:

    | Dataset   | Eigen Decomposition Time per Sample| Total Processing Time  |
    |-----------|------------------------------------|------------------------|
    | Cylinder  | 0.026 s                            | 27.21 s                |
    | Airfoil   | 0.204 s                            | 218.34 s               |
    | Plate     | 0.016 s                            | 19.19 s                |
    | Flag      | 0.020 s                            | 21.82 s                |

    While we provide a precomputation cost table for clarity, this data processing only occurs once and can be considered part of the dataset generation process.

---

### Author Response · Authors · 2024-11-21
**General response Part 2.**

5. **[Experiment: ablation in FlagSimple]**[Reviewer wc4g]:

    | Measurements    | Without HPA and GFL | HPA only | GFL only | HPA + GFL (Ours) |
    |-----------------|---------------------|----------|----------|------------------|
    | RMSE-1 [1e-2]   | 6.47E-2             | 1.51E-2  | 2.29E-2  | **1.12E-2**      |
    | RMSE-50 [1e-2]  | 2.29                | 1.97     | 2.03     | **1.87**         |
    | RMSE-all [1e-2] | 2.45                | 2.16     | 2.21     | **2.05**         |

    The ablation study focusing on the FlagSimple dataset.

6. **[Experiment: Additional Baslines Comparison]**[Reviewer L6SR, Reviewer YBDW, Reviewer ZnqH]:

    | Measurements    | Dataset   | Ours (MPT)         |Mesh Transformer[1]| HCMT[3]          |Graph MLP-MIXER[4]|
    |-----------------|-----------|--------------------|-------------------|------------------|------------------|
    | RMSE-1 [1E-2]   | Cylinder  | **2.03 E-01**      | 3.05 E-01         | -                | 4.12 E-01        |
    |                 | Airfoil   | **2.61 E+02**      | 2.96 E+02         | -                | 3.05 E+02        |
    |                 | Plate     | **1.00 E-02**      | 2.38 E-02         | 1.15 E-02        | 3.28 E-02        |
    |                 | Flag      | **1.12 E-02**      | 4.73 E-02         | -                | 6.89 E-02        |
    | RMSE-50 [1E-2]  | Cylinder  | **6.32 E-01**      | 1.07              | -                | 4.61             |
    |                 | Airfoil   | **4.08 E+02**      | 5.21 E+02         | -                | 6.32 E+02        |
    |                 | Plate     | **9.25 E-02**      | 1.30 E-01         | 9.68 E-01        | 6.19 E-01        |
    |                 | Flag      | **1.87**           | 2.04              | -                | 3.90             |
    | RMSE-all [1E-2] | Cylinder  | **3.78**           | 4.26              | -                | 2.05 E+01        |
    |                 | Airfoil   | **1.64 E+03**      | 2.00 E+03         | -                | 3.97 E+03        |
    |                 | Plate     | **1.09**           | 1.28              | 1.15             | 8.22             |
    |                 | Flag      | **2.05**           | 2.32              | -                | 7.25             |

    We additionally evaluated ours MPT against the Mesh Transformer[1], HCMT[3], and Graph MLP-MIXER[4] across four datasets.

7. **[Experiment: Additional Dataset Test]**[Reviewer L6SR]:

    | Measurements    |Ours (MPT)    | MGN       | BSMS        | TIE       |
    |-----------------|--------------|-----------|-------------|-----------|
    | RMSE-1          | **1.08 E-01**| 2.25 E-01 | 1.87 E-01   | 1.62 E-01 |
    | RMSE-50         | **2.57 E-01**| 6.03 E-01 | 5.34 E-01   | 5.71 E-01 |
    | RMSE-all (100)  | **4.63 E-01**| 9.25 E-01 | 8.36 E-01   | 8.49 E-01 |

    We additionally conducted an experiment on the Dam Problem from the 2023 CFDBench [2] benchmark.

8. **[Experiment: Runtime Consistency Performance]**[Reviewer L6SR]:

    | Measurements    | Ours (MPT)    | MGN (128)  | MGN (256)  | BSMS (128)  | BSMS (256)  | TIE (128)  | TIE (256)  |
    |-----------------|---------------|------------|------------|-------------|-------------|------------|------------|
    | RMSE-1          | **1.00 E-02** | 2.69 E-02  | 2.14 E-02  | 2.83 E-02   | 2.65 E-02   | 3.56 E-02  | 3.21 E-02  |
    | RMSE-50         | **9.25 E-02** | 1.73 E-01  | 1.62 E-01  | 2.81 E-01   | 2.43 E-01   | 3.61 E-01  | 3.37 E-01  |
    | RMSE-all (100)  | **1.09**      | 1.61       | 1.52       | 4.52        | 4.34        | 9.62       | 9.05       |


    We trained and tested MGN, BSMS, and TIE with a hidden layer dimension of 256 on the DeformingPlate dataset, resulting in a training budget slightly higher than our method.

9. **[Experiment: GFL Robustness Performance]**[Reviewer ZnqH]:

    | Edge Removal Percentage  | High Energy Mean          | Low Energy Mean           | Energy Ratio (High/Low)        |
    |--------------------------|---------------------------|---------------------------|--------------------------------|
    | 1%                       | 3.388615 ± 0.014727       | 3.391265 ± 0.014988       | 1.000784 ± 0.000418            |
    | 5%                       | 3.387550 ± 0.014215       | 3.389730 ± 0.014485       | 1.000643 ± 0.000392            |
    | 10%                      | 3.387155 ± 0.013993       | 3.389425 ± 0.014275       | 1.000670 ± 0.000385            |


    Energy changes after randomly removing different proportions of edges on the Airfoil dataset.

[1] Eagle: Large-scale learning of turbulent fluid dynamics with mesh transformers

[2] CFDBench: A Comprehensive Benchmark for Machine Learning Methods in Fluid Dynamics

[3] Learning Flexible Body Collision Dynamics with Hierarchical Contact Mesh Transformer

[4] A GENERALIZATION OF VIT/MLP-MIXER TO GRAPHS

---

### Author Response · Authors · 2024-11-27
**The revised paper has been uploaded**

We appreciate the time and effort of all reviewers and the AC in reviewing our work, as well as their insightful comments, which have significantly strengthened it.

# Updates and Changes
## Introduction
- Added citations of **The Universal Physics Transformers**, **Poseidon: Efficient Foundation Models for PDEs**, **GraphGPS**, **GAT**, and **Graph MLP-Mixer**.
- More clearly defined the key challenges our Historical Message-Passing Attention (HMPA) addresses:
  - **Mitigating over-squashing**
  - **Capturing fine-grained details**
  - **Scaling linearly with node count**

## Problem Formulation and Preliminaries
- Revised Section 3.1 (*Problem Formulation*) for improved clarity. Corresponding symbols were updated, and the assumption that graph topology remains constant over time was removed.
- Clarified ambiguous descriptions in Section 3.2.1 (*Mathematical Definitions*), particularly regarding the adjacency matrix.

## Method
- Removed Sections 4.2.1 (*Gradient Computation in HPA*) and 4.2.3 (*Dynamic Feature Importance*).
- Added Section 4.2.1 (*Complexity Analysis*).
- Refined and improved expressions for better clarity.

## Experiment
- **[Experiment: Precomputation Cost]** was added as part of Section 5.3 (*Precomputation Cost*), along with Table 2 and corresponding analysis.
- **[Experiment: Ablation in FlagSimple]** was added to Section 5.4 (*Ablation Studies*), included as Table 3.
- **[Experiment: Additional Baselines Comparison]** was incorporated into Section 5.2 (*Comparison with Baselines*), included as Table 1 with corresponding revisions. While we evaluated HCMT [1] during the rebuttal phase, HCMT was tested only on the *DeformingPlate* dataset because its **CMT** and **HMT** modules were specifically designed for collision cases. Therefore, HCMT is not included in the final paper.
- **[Experiment: Additional Dataset Test]** was added to *Appendix G (Performance of HMIT on Dam Flow)* as Table 9 with corresponding analysis.
- **[Experiment: Runtime Consistency Performance]** was not included in the paper. During our response to Reviewer ZnqH, we identified further optimization opportunities, making Table 1 more comparable in terms of parameter count, runtime, and memory consumption. Consequently, comparisons with baselines using hidden layers of 256 dimensions were no longer necessary.
- **[Experiment: GFL Robustness Performance]** was added to *Appendix B (Experimental Analysis of Graph Fourier Loss)* as Table 4, along with corresponding analysis.
- The new **[Experiment: Memory Consumption and Computational Speed]** was added to *Appendix F (Memory Consumption and Computational Speed)* as Table 8 with corresponding updates.
-  We have added a clarification regarding the fact that world edges do not participate in the Laplacian eigendecomposition in the paragraph 'Noise Injection and World Edge Radius Settings' in Appendix E (Model Details).

## Visualizations
- **Figure 1**: Corrected the label error and clarified the abbreviation.
- **Figure 3(a)**: Corrected the label error.
- **Appendix H**: Updated Rollout Visualizations, now including comparisons between Ground Truth and ours HMIT.


[1] Learning Flexible Body Collision Dynamics with Hierarchical Contact Mesh Transformer

---

### Meta-Review · Area_Chair_XmRm · 2024-12-17

**Metareview:**

This paper introduces a novel hybrid graph neural network transformer (Message Passing Transformer, MPT). At the technical level, the main contributions consist in using a scaled Hadamard product attention and a new loss function employing the graph Fourier transform. Numerical results on Eulerian and Lagrangian dynamics show promising improvements over the state of the art.

The reviewers have appreciated the novelty in the approach and the improvements demonstrated in experiments. However, several issues related to the benefits of the method in comparison with related work and to its computational costs have been raised. In response, the authors have provided a rather thorough rebuttal, presenting a number of additional experiments. This was appreciated by the reviewers that have updated their scores, making the paper borderline.

As the paper has changed significantly (even at the level of the abstract and of the description of the main contributions), my opinion -- which is shared with various reviewers -- is that it still needs another round of thorough reviewing to ensure a proper evaluation. It is in fact not entirely clear that all the main objections of the reviewers have been positively resolved. For this reason, I suggest a rejection at this stage but would like to encourage the authors to resubmit an improved version to a future venue.

**Additional Comments On Reviewer Discussion:**

The authors have uploaded a detailed rebuttal and discussed with the reviewers. However, given the amount of necessary changes to the paper, there is no clear consensus towards accepting the paper.

---

### Decision · Program_Chairs · 2025-01-22

Reject